# Plasma membrane LAT activation precedes vesicular recruitment defining two phases of early T-cell activation

Lakshmi Balagopalan[1], Jason Yi[1], Tiffany Nguyen[1], Katherine M. McIntire[1], Adam S. Harned[2,3], Kedar Narayan[2,3] & Lawrence E. Samelson[1]

The relative importance of plasma membrane-localized LAT versus vesicular LAT for microcluster formation and T-cell receptor (TCR) activation is unclear. Here, we show the sequence of events in LAT microcluster formation and vesicle delivery, using lattice light sheet microscopy to image a T cell from the earliest point of activation. A kinetic lag occurs between LAT microcluster formation and vesicular pool recruitment to the synapse. Correlative 3D light and electron microscopy show an absence of vesicles at microclusters at early times, but an abundance of vesicles as activation proceeds. Using TIRF-SIM to look at the activated T-cell surface with high resolution, we capture directed vesicle movement between microclusters on microtubules. We propose a model in which cell surface LAT is recruited rapidly and phosphorylated at sites of T-cell activation, while the vesicular pool is subsequently recruited and dynamically interacts with microclusters.

[1] Laboratory of Cellular and Molecular Biology, Center for Cancer Research, National Cancer Institute, National Institutes of Health, 37 Convent Drive, Bethesda, MD 20892, USA. [2] Center for Molecular Microscopy, Center for Cancer Research, National Cancer Institute, National Institutes of Health, 8560 Progress Drive, Frederick, MD 21701, USA. [3] Cancer Research Technology Program, Frederick National Laboratory for Cancer Research, Frederick, Maryland, USA. Correspondence and requests for materials should be addressed to L.B. (email: balagopl@mail.nih.gov) or to L.E.S. (email: samelsonl@mail.nih.gov)

T cells express T-cell receptors (TCR) on their surface that bind and detect antigens. Engagement of the TCR by a peptide-bound major histocompatibility complex (pMHC) molecule results in the phosphorylation of the signal transducing CD3 and TCRζ chains by the Src family kinase Lck. ZAP-70, a second tyrosine kinase, is recruited from the cytosol to the phosphorylated receptor and in turn is phosphorylated and fully activated by Lck[1]. Activated ZAP-70 phosphorylates linker for activation of T cells (LAT), a transmembrane adapter protein essential for T-cell signaling. Several studies in cell lines and mice have established the central importance of LAT in TCR signaling. The phosphorylated tyrosines on LAT are nucleation sites for adapters and important signaling complexes that together mediate T-cell activation[2].

Microscopy studies have identified that T-cell engagement results in the rapid formation of microclusters containing many signaling molecules[3, 4]. Microclusters form within seconds of TCR engagement and are the basic signaling units required for T-cell activation. However, the critical sequence of events by which T cells establish signaling microclusters is unclear. LAT is localized at the plasma membrane and also in intracellular vesicles in resting and stimulated cells[5, 6]. The relative importance of plasma membrane-localized LAT versus vesicular LAT for TCR signal transduction is a subject of active debate. There are two very different points of view regarding which LAT pool is recruited to microclusters and participates in TCR signaling. In one model, direct recruitment of cell surface LAT to microclusters is critical for T-cell activation[7–10], while in another model, vesicular, but not cell surface LAT, is essential[11–14].

The evidence for the first model involving plasma membrane-resident LAT comes from transmission electron microscopy (TEM) and super-resolution photoactivated localization microscopy (PALM) studies that propose that cell surface LAT is pre-clustered at the plasma membrane and cluster sizes increase upon T-cell stimulation[7–9]. Using chimeric LAT with an extracellular tag, we previously provided evidence that cell surface LAT is efficiently recruited to microclusters, becomes phosphorylated, and propagates signals downstream of the TCR[10]. The evidence for the second model and the role of vesicular LAT in T-cell activation came initially from a study that demonstrated that a substantial fraction of LAT was present in subsynaptic vesicles and the observation that LAT phosphorylation coincided with subsynaptic vesicle interaction with microclusters[11]. Williamson et al.[12] using super-resolution microscopy reported that pre-existing LAT domains at the plasma membrane did not get phosphorylated or recruited to TCR activation sites. In another study, vesicular LAT was shown to be localized to the calcium-sensitive Rab27a–Rab37–VAMP7 exocytic compartment and an artificial increase of intracellular calcium in cells led to the release of vesicular LAT to the PM[13]. Interfering with LAT release from vesicular compartments by silencing vesicular fusion machinery such as the calcium sensor synaptotagmin7, or the vesicular SNARE VAMP7, resulted in decreased LAT phosphorylation and IL-2 production[13, 14]. From these results, it was proposed that calcium-dependent exocytosis of vesicular LAT is the primary mechanism by which LAT is recruited to microclusters, phosphorylated, and propagates downstream signals in an activated T cell. Thus, two different models by which LAT gets activated in a T cell currently exist. Distinguishing between these models is central to understanding how immune responses are triggered and maintained.

To interrogate the relationship between plasma membrane and vesicular pools and how they contribute to microcluster formation, we simultaneously imaged LAT and a vesicular marker VAMP7, which partially colocalizes with intracellular LAT, from the instant a T cell becomes activated. However, conventional 4D imaging technologies such as spinning-disk microscopy induce photobleaching and image at speeds that are too slow to capture microcluster dynamics. To overcome these hurdles, we used a faster, minimally phototoxic 4D microscopy technique, lattice light sheet microscopy (LLSM)[15]. Because two-color lattice light sheet imaging enabled simultaneous identification of LAT microclusters and intracellular vesicular pools at temporal acquisition rates fast enough to image microcluster formation, we could interrogate the relationship between these two cellular pools. We observe a kinetic lag between LAT microcluster formation and vesicular pool recruitment to the synapse. Given that vesicles come close to the activated T-cell surface at later time points, we imaged this region with higher spatial and temporal resolution using total internal reflection fluorescence- simulated interference microscopy (TIRF-SIM) in which the structured illumination pattern is confined to the TIRF plane, thus merging the strength of TIRF and SIM microscopy[16, 17]. Using TIRF-SIM, we capture directed vesicle movement on microtubules between microclusters where they exhibit increased dwell times. Using correlative 3D light and focused ion beam scanning electron microscopy (FIB-SEM) to locate microclusters in the context of cellular ultrastructure, we observe an absence of vesicles at microclusters at early times after stimulation, but an abundance of vesicles at the stimulated surface as activation proceeded. We propose a model in which cell surface LAT is rapidly recruited and phosphorylated at sites of T-cell activation, and the vesicular pool is recruited subsequently. Once recruited, vesicles travel on microtubules and dynamically interact with microclusters.

## Results

**LAT microclusters form in the absence of vesicles.** Vesicular LAT partially colocalizes with the calcium-sensitive Rab27a–Rab37–VAMP7 exocytic compartment[13]. It has been proposed that calcium influx initiated by TCR triggering causes calcium-dependent exocytosis of vesicular LAT, thus allowing for LAT recruitment to microclusters and phosphorylation[13]. In contrast, cells in which TCR-induced calcium elevations were inhibited using calcium chelators, showed normal LAT signaling complex formation upon TCR activation[18], arguing against a role for calcium flux at the initiation of T-cell signaling. To sort out this discrepancy, we looked at microcluster formation in cells in which TCR-induced calcium elevations were completely blocked. We performed spreading assays in which cells were dropped on coverslips coated with antibody that is stimulatory to the TCR. Cells were dropped in media buffered with EGTA, a calcium chelator, and pretreated with BAPTA-AM, a membrane-permeant calcium chelator. These conditions prevent all intracellular calcium elevations (Supplementary Figure 1a). If calcium-induced exocytosis is required for microcluster formation, LAT microclusters should be severely decreased in BAPTA-treated cells. In contrast, after both 2 and 5 min of stimulation, we observed robust phosphotyrosine and phospho-LAT clusters in cells in which calcium flux was completely inhibited (Fig. 1a). Quantification of phospho-LAT cluster size and intensity revealed no differences in phospho-LAT clusters at early times, but a slight increase in the size and brightness of these clusters in BAPTA pretreated cells at later time points (Fig. 1b, c), while quantification of phosphotyrosine cluster intensity showed a slight increase at both early and late times (Supplementary Figure 1b). These results clearly indicate that calcium influx is not required for LAT microcluster formation and argue that calcium-mediated exocytosis of vesicles is not required for LAT microcluster formation.

In addition to microcluster formation, TCR stimulation results in a series of signaling events that direct reorientation of the microtubule-organizing center (MTOC) toward the activated T-

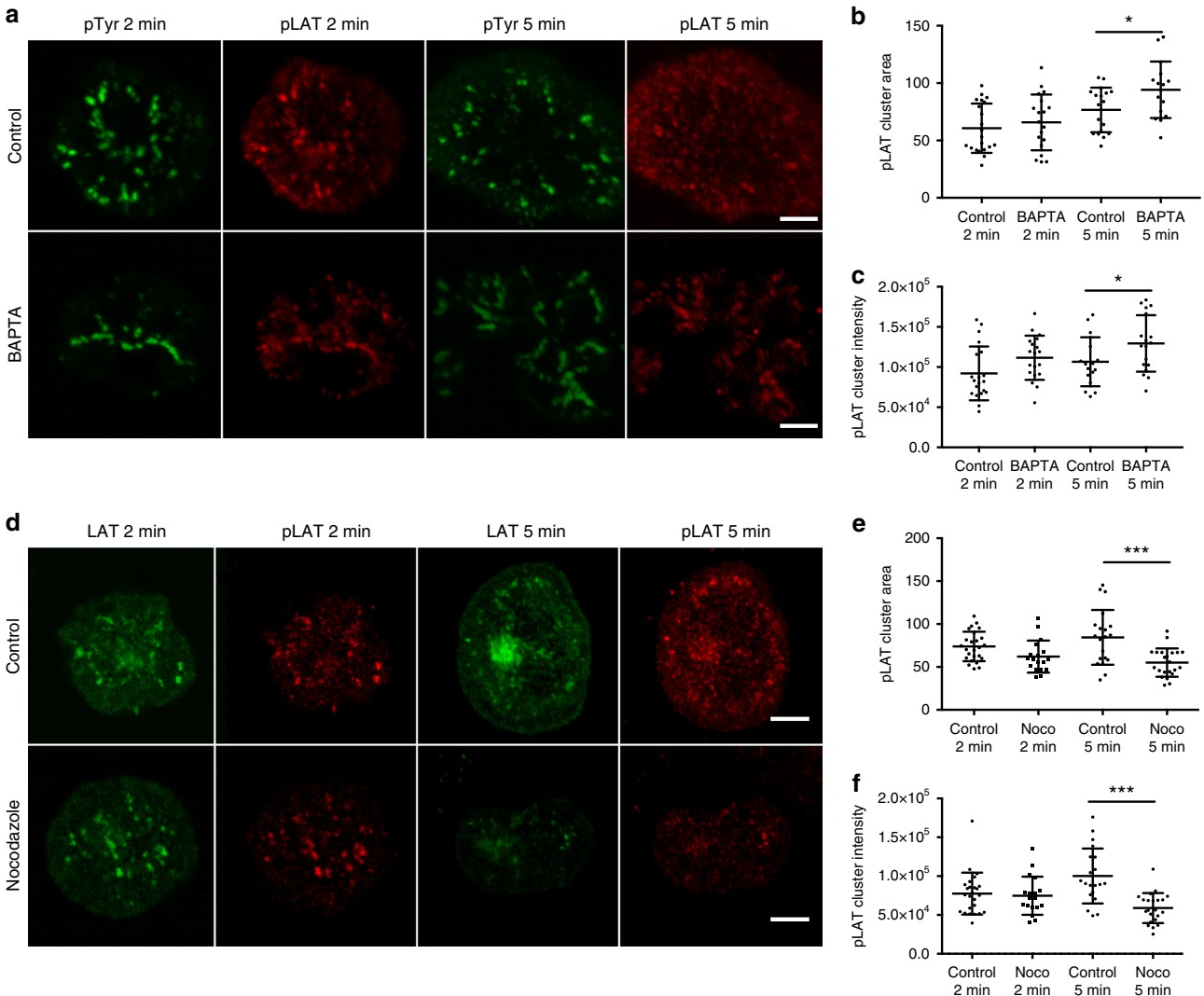

**Fig. 1** BAPTA and nocodazole inhibition do not affect LAT microcluster formation. **a** Jurkat cells were treated with DMSO and dropped onto stimulatory coverslips in a medium containing DMSO (control) or treated with BAPTA and dropped onto stimulatory coverslips in a medium containing EGTA (BAPTA), and fixed 2 and 5 min after dropping. Cells were then permeabilized and immunostained with pTyr and pLAT. Scale bar equals 2 μm. **b** Cluster area and **c** cluster intensity of pLAT microclusters were quantified. **d** Jurkat cells transfected with LAT-ruby were treated with DMSO and dropped onto stimulatory coverslips in a medium containing DMSO (control) or treated with nocodazole and dropped onto coverslips in a medium containing nocodazole (nocodazole) and fixed 2 and 5 min after dropping. Cells were then permeabilized and immunostained with pLAT. Scale bar equals 2 μm. **e** Cluster area and **f** cluster intensity of pLAT microclusters were quantified. See also Supplementary Figure 1

cell interface[19]. Such MTOC polarization to the immunological synapse (IS) causes the reorientation of various vesicular compartments to the subsynaptic zone. VAMP7, a SNARE that is important for microcluster formation, partially colocalizes with intracellular LAT[13, 14] and VAMP7-containing vesicles have been shown to move on microtubules to the cell periphery[20]. To interfere with microtubule-mediated vesicle delivery, we treated cells with nocodazole to depolymerize the microtubule cytoskeleton. Control T cells treated with DMSO displayed robust recruitment of VAMP7-containing vesicles to the activating surface, while treatment with nocodazole resulted in a small, but significant, defect in VAMP7-positive vesicle reorientation to the activation surface (Supplementary Figure 1c). A concentration of nocodazole was chosen to not inhibit TCR activation[21], and the disruption of microtubules by nocodazole treatment was confirmed (data not shown). Interestingly, in nocodazole-treated cells, phospho-LAT microcluster formation was intact (Fig. 1d). Quantification of phospho-LAT and total LAT cluster

size and intensity in nocodazole-treated cells appeared similar at early time points, but decreased at later time points. (Fig. 1e, f; Supplementary Figure 1d). These results indicate that neither vesicle recruitment nor calcium-mediated exocytosis are required for LAT microcluster formation.

**Microclusters precede vesicles at sites of activation.** The cellular origin of LAT microclusters remains controversial due to imaging studies that have reported conflicting results regarding the contributions of cell surface versus vesicular LAT pools[3, 5, 10–14, 22]. To understand the temporal relationships between membrane and vesicular pools that lead to microcluster formation, we simultaneously imaged the transmembrane adapter LAT and VAMP7 in T cells using total internal reflection fluorescence (TIRF) microscopy from the instant a T cell becomes activated. This technique allows visualization of vesicles within 100–200 nm of the stimulated cell surface. Jurkat T cells were cotransfected with GFP-VAMP7

and LAT-RFP and dropped on stimulatory coverslips. LAT microclusters formed within the first few seconds of contact with the stimulatory coverslip. As the cell spreads, LAT clusters formed in areas of cell contact with the coverslip, and once the cell spreads, the majority of LAT in large microclusters dissipated, though smaller microclusters persisted. Importantly, the earliest time VAMP7-positive vesicles appeared in the TIRF field was 45–60 s after initiation of imaging when cell spreading was well underway, followed by rapid recruitment of additional VAMP7-positive vesicles (Fig. 2a–c; Supplementary Movie 1). To evaluate whether LAT cluster dynamics reflected activated LAT, Grb2, an adapter molecule that is recruited to phosphorylated tyrosines on LAT, and thus serves as an activated LAT reporter, was imaged simultaneously with VAMP7. Similar to LAT, Grb2 microclusters formed upon cell contact with the activating surface, followed by the earliest VAMP7 recruitment at around 45–60 s and subsequent recruitment of additional VAMP7-positive vesicles (Fig. 2d; Supplementary Movie 2). In summary, LAT and Grb2 microcluster formation preceded VAMP7 recruitment to the activated T-cell surface, indicating that vesicular LAT does not significantly contribute to the initiation of LAT microcluster formation.

**LAT and VAMP7 have different spatial and temporal dynamics.** Though these TIRF experiments allowed us to define the temporal relationship between LAT-Grb2 microclusters and VAMP7-positive vesicles closely apposed to the activated T-cell surface, assessment of vesicle mobility in three dimensions (3D) in the cell volume was lost. To rapidly image the 3D volume of the cell, we sought a faster, minimally phototoxic 4D microscopy technique, LLSM[15], that has been previously used to image granule secretion in cytotoxic T cells[23]. Using LLSM, we obtained a 15-fold increase in temporal resolution and a threefold increase in optical sectioning. Because two-color lattice light sheet imaging

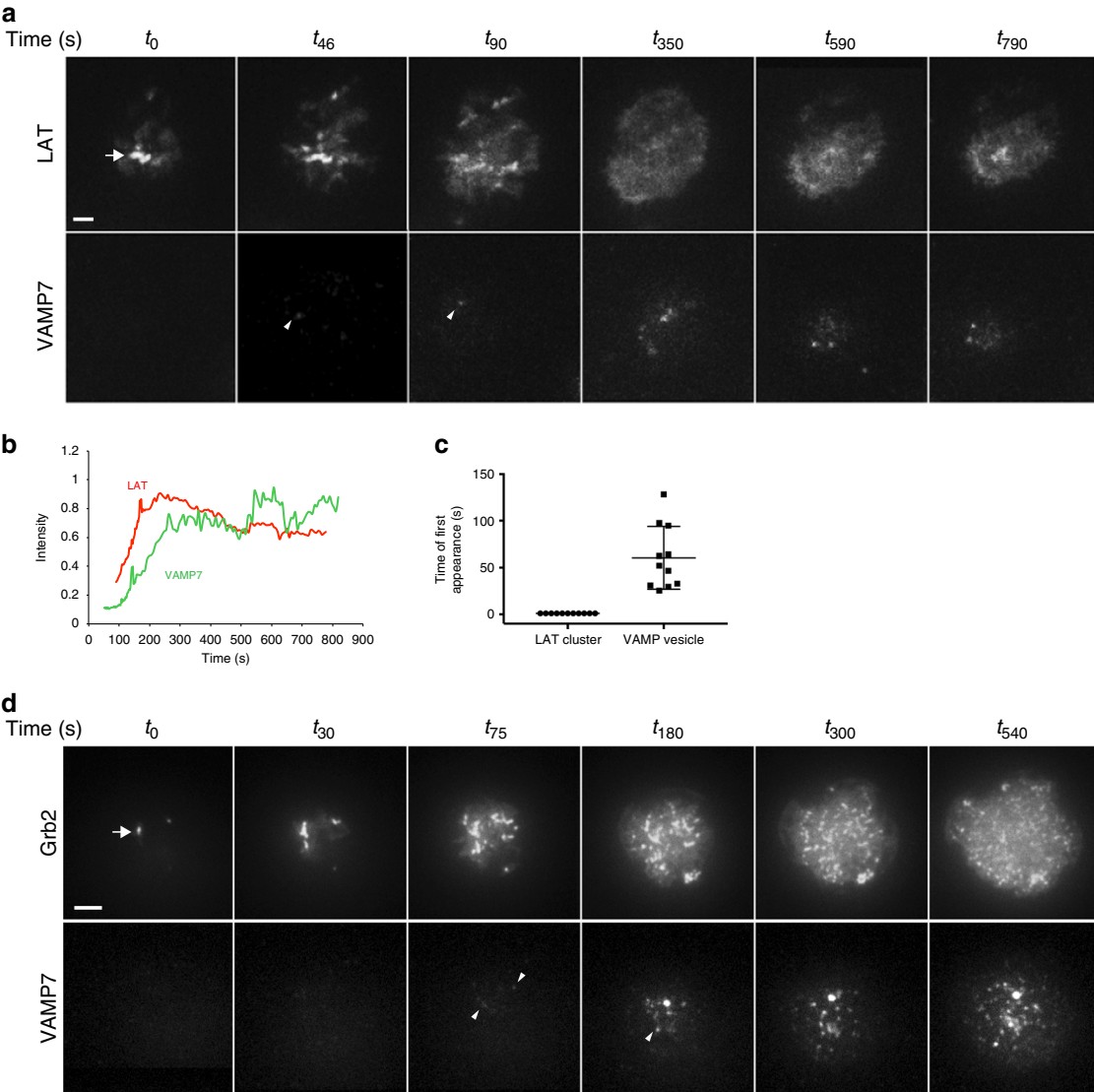

**Fig. 2** Microcluster formation precedes VAMP7 recruitment as visualized by TIRF microscopy. **a** Jurkat cells transfected with LAT-RFP and GFP-VAMP7 were dropped onto stimulatory coverslips and imaged by TIRF microscopy. **b** Intensity profile of LAT-RFP and GFP-VAMP7 over time. **c** Time of appearance of LAT-RFP or GFP-VAMP7 signal in TIRF field ($n = 12$ cells, three independent experiments). **d** Jurkat cells transfected with Grb2-apple and emerald-VAMP7 were dropped onto stimulatory coverslips and imaged by TIRF microscopy ($n = 10$ cells, three independent experiments). **a**, **d** Top panels show LAT and Grb2, respectively, and arrows point to early microclusters; bottom panels show VAMP7, and arrowheads point to vesicles touching down. Scale bars = 2 μm

enabled simultaneous identification of LAT microclusters and intracellular vesicular pools at temporal acquisition rates fast enough to image microcluster formation, we could interrogate the relationship between these two cellular pools.

Jurkat cells were cotransfected with LAT-neon green and Halo-VAMP7. In unstimulated cells, LAT was observed at the plasma membrane and in vesicles as previously described[5]. VAMP7 was observed in two distinct pools of intracellular vesicles, and vesicular LAT overlapped partially with one of these pools in unstimulated cells (Supplementary Figure 2a, Supplementary Movie 3). When cells expressing LAT and VAMP7 were dropped onto stimulatory coverslips, LAT microclusters were observed at the activating planar surface at times when the VAMP7 pool that partially colocalizes with vesicular LAT was several microns away from the activating surface (Fig. 3a top panel, the white arrow at $t_0$ indicates a pool of VAMP7 that partially colocalizes with intracellular LAT). Microtubule polarization toward the IS causes the reorientation of various vesicular compartments to the subsynaptic zone[19]. Consistent with such a reorientation, the pool of VAMP7 that colocalizes with LAT and is likely associated with the MTOC, was recruited to the activating surface within 30–45 s of initiation of imaging (Fig. 3a top panel, $t_{44}$; Supplementary Movie 4). This is clearly shown in the en face views of the stimulated surface (Fig. 3a bottom panel, arrowheads at all times indicate LAT microclusters and arrows at $t_{44}$, $t_{132}$, and $t_{191}$ indicate VAMP7 vesicles recruited to the synapse). Following recruitment to the synapse, very dynamic VAMP7 vesicular movement and intermixing of vesicles from the two pools was observed.

To gain further insight into the relationship between VAMP7-positive vesicles and LAT microclusters, we used fluorescence thresholding to render surfaces of the VAMP7 pools and LAT clusters (Fig. 3b left panel). Within the VAMP7 surfaces, the individual VAMP7 vesicles were assigned as spots that were then followed over the time-course of the movie (Fig. 3b middle panel). Spot distances from the LAT microcluster surface were calculated to generate a distance map of vesicles to clusters (Fig. 3b right panel; Supplementary Movie 5). Plotting the distance of VAMP7 vesicles to LAT microclusters over time shows that there are no vesicles in contact with LAT microclusters for the first 45 s, after which the vesicles remain in proximity and interact with the clusters (Fig. 3c). Similar to the TIRF data described above, the first wave of prominent LAT microclusters is already decreasing in fluorescence intensity when the VAMP7-positive vesicles make contact with the activation surface (Supplementary Figure 2b), suggesting that the first wave of LAT microclusters originates from the cell surface pool of LAT and is not dependent on vesicular compartments.

**LAT accumulates at the IS before VAMP7 recruitment**. To evaluate the relationship between cell surface and vesicular pools in the more physiological context of an immune synapse, Jurkat T cells cotransfected with LAT-neon green and Halo-VAMP7 were incubated with SEE-pulsed Raji B cells to represent antigen-pulsed APCs, and synapse formation was imaged from the initiation of contact between the two cells. Upon interaction of the Jurkat T cell with the Raji B cell, dense LAT clusters were immediately observed at the interface of the two cells (Fig. 3d, $t_0$). These clusters represent activated LAT as Grb2 colocalizes with LAT clusters (Supplementary Movie 6). Following cluster formation, the MTOC reoriented to the IS along with MTOC-associated vesicular compartments visualized by VAMP7 (Fig. 3d top panel, $t_{231}$). Thus, LAT microcluster formation preceded vesicle recruitment in an immune synapse, consistent with what we observed on stimulatory coverslips. This was strikingly observed when a Jurkat cell, which had already formed a synapse

with a B cell (synapse 1), initiated contact with another B cell in close proximity. Robust LAT microcluster formation was observed at this second synapse (synapse 2), even though the vesicular compartments were oriented toward synapse 1 (Fig. 3d, $t_{231}$; Supplementary Movie 7). This is clearly shown in the en face views of synapses 1 and 2 (Fig. 3d middle and bottom panels). These results demonstrate that LAT microclusters at the immune synapse are not dependent on the proximity of vesicular compartments. Together, the above-described results suggest a major role for cell surface LAT in initiating T-cell activation prior to vesicular pool recruitment to the synapse.

**Microcluster ultrastructure shows two phases of activation**. To better understand the ultrastructural context of microcluster formation and vesicular recruitment, we used correlative 3D light and electron microscopy (CLEM) to examine activated T cells at both early and late time points. Jurkat cells were transfected with emerald-VAMP7 and dropped onto stimulatory, alphanumerically coded, gridded coverslips, fixed at different times after activation, and labeled for nuclei, plasma membrane, and phosphorylated LAT microclusters. Activated T cells were imaged by confocal imaging, after which the samples were processed for EM, where they were unambiguously located using the grid pattern, ensuring that the same cells were imaged by both LM and EM. However, the visualization of ~100-nm-sized microclusters and their differentiation from other membranous structures in these cells requires the ability to image large volumes in 3D and at nanoscale resolutions. A 3D EM technique, FIB-SEM[24] was thus used to locate and image activated T cells in their entirety. Using previously reported advances[25] and protocol modifications, we executed automated runs of synchronized FIB milling and SEM imaging cycles to generate large stacks of ultrastructural images that encompassed individual T cells, with minimal imaging artifacts. These stacks were computationally converted into high-quality image volumes at ~10 nm isotropic 3D pixel sampling. Increased image quality of the membranous structures in the surface-proximal region of the cells, coupled with high resolutions afforded by FIB-SEM and correlation with confocal image volumes, allowed for the ultrastructural identification of features imaged by LM in these cells. It must be noted that while we were able to correlate and visualize subvolumes in the FIB-SEM datasets corresponding to signals within the PSF of the LM data, the limiting spatial resolutions of LM precluded the specific identification of individual vesicles as VAMP7 or LAT positive.

At early time points after activation, pLAT microclusters were detected at the activated T-cell surface (Fig. 4a, ROI 1), while VAMP7-positive vesicles were located further above in the cell volume (Fig. 4a, ROI 2), consistent with LLSM live cell imaging results described above. FIB-SEM images revealed very few structures identifiable as vesicles close to the activated T-cell membrane, though the nuclear and plasma membranes were clearly visible (Fig. 4c–e, FIB-SEM slices Supplementary Movie 8 and segmented volume Supplementary Movie 9, segmented volume with membranes shown in Supplementary Figure 3a). By contrast, the ultrastructure of VAMP7-positive ROI 2 clearly detected several vesicles approximately 100 nm in diameter (Fig. 4f–h, FIB-SEM slices Supplementary Movie 10 and segmented volume Supplementary Movie 11, segmented volume with membranes shown in Supplementary Figure 3b and c), thus establishing that these vesicles were identifiable by FIB-SEM. At later times after activation, both light and FIB-SEM images revealed changes in cell morphology that were characteristic of stimulation. The cells had spread and VAMP7-positive vesicles were recruited to the activated T-cell surface in close proximity to pLAT microclusters (Fig. 4i, ROI 1). Examples of pLAT clusters

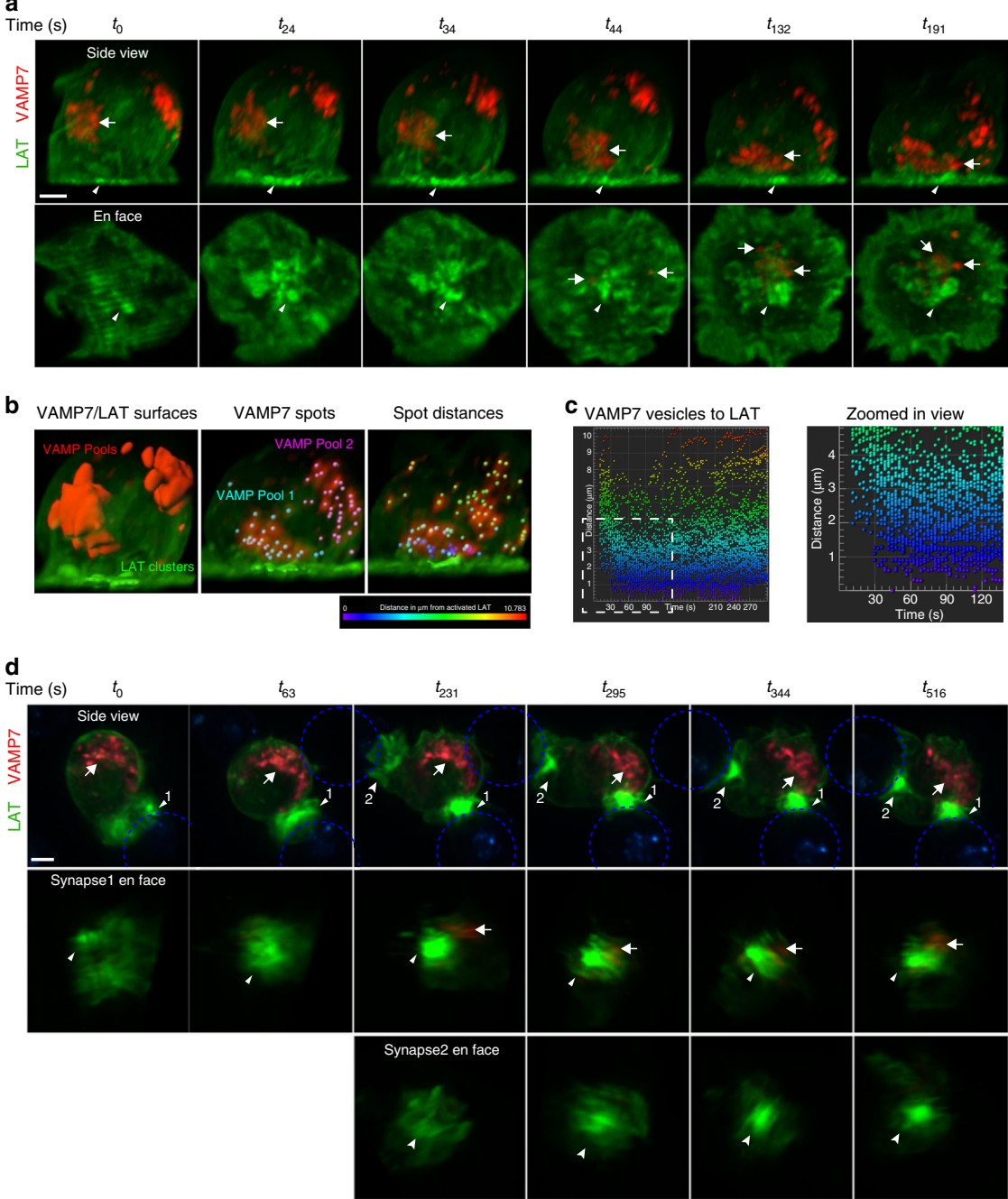

**Fig. 3** Spatial and temporal dynamics of LAT and VAMP7 visualized by lattice light sheet microscopy. **a** Jurkat cells transfected with LAT-neon green (green) and Halo-VAMP7 (red) were dropped onto stimulatory coverslips and imaged by lattice light sheet microscopy soon after activation. $t_0$ indicates the first time point of image collection. The top panel shows the side view of the cell. The arrowhead indicates LAT microclusters and the arrow indicates VAMP7 vesicles that colocalize with vesicular LAT. Scale bar equals 2 μm. The bottom panel shows en face view of the activated surface ($n = 5$ cells, three independent experiments). **b** The left panel shows surfaces of LAT clusters (green) and VAMP7 vesicles (red). The middle panel shows individual vesicles marked as spots (cyan and pink) within VAMP7 surfaces. The panel to the right shows distances of VAMP7 spots from LAT clusters followed over time. The vesicles are color-coded to indicate distance in μm from activated LAT. **c** The panel to the left shows the graph of distances of individual VAMP7 vesicles from LAT clusters over time. Vesicles are color-coded to indicate distance in μm from activated LAT as in **b**. To the right is the zoomed-in view of the white boxed region indicated in the graph. **d** Jurkat cells transfected with LAT-neon green (green) and Halo-VAMP7 (red) were incubated with SEE-coated Raji B cells (blue) and immune synapse formation was visualized by lattice light sheet microscopy. $t_0$ indicates the first time point of image collection. The top panel shows the side view of the cell. The arrowhead indicates LAT aggregated at immune synapse 1, and the arrow indicates VAMP7 vesicles. At $t_{231}$, the T cells begin to interact with a second Raji B cell. The second synapse is indicated with an indented arrowhead (synapse 2). Scale bar equals 2 μm. The middle panel shows en face view of synapse 1 showing LAT accumulation at the immune synapse before the appearance of VAMP7 vesicles at $t_{231}$. The bottom panel shows en face view of synapse 2 ($n = 5$ cells, three independent experiments). See also Supplementary Figure 2

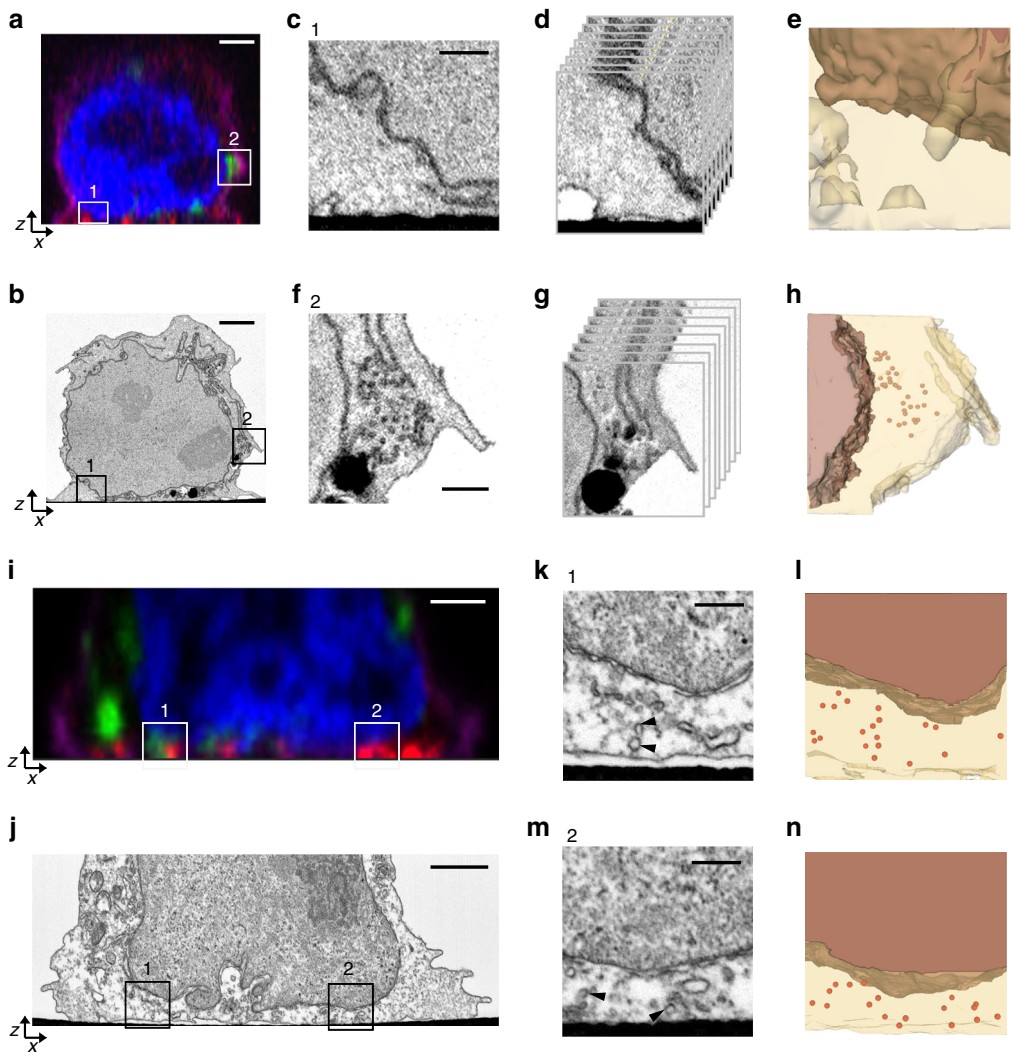

**Fig. 4** Correlative 3D light and FIB-SEM of activated T cells. Jurkat cells transfected with emerald-VAMP7 (green) were dropped onto stimulatory coverslips and fixed after 2 min **a–h** or 5 min **i–n** of activation and immunostained for pLAT (red), nucleus (blue), and plasma membrane (purple). **a** Images were collected of the whole cell activated for 2 min using confocal microscopy. Box 1 indicates a region with a pLAT microcluster and Box 2 indicates a region with VAMP7 vesicles. **b** Focused ion beam scanning electron microscopy (FIB-SEM) images were subsequently collected from the same samples. Scale bars for **a** and **b** equal 2 μm. **c**, **f** Zoomed-in FIB-SEM image of ROI 1 and ROI 2, respectively, showing a single slice. Scale bars for **c** and **f** equal 0.5 μm. **d**, **g** Zoomed-in FIB-SEM stacks corresponding to ROI 1 and 2, respectively. Corresponding FIB-SEM stacks are in Supplementary Movies 8 and 10, respectively. **e**, **h** Segmented FIB-SEM volumes corresponding to ROI 1 and 2, respectively; correspond to Supplementary Movies 9 and 11, respectively. **i** Images were collected through the whole cell activated for 5 min using confocal microscopy. Box 1 indicates a region with a pLAT microcluster in proximity of VAMP7 signal and Box 2 indicates a pLAT microcluster devoid of VAMP7 signal. **j** Focused ion beam scanning electron microscopy (FIB-SEM) images were subsequently collected from the same samples. Scale bars for **i** and **j** equal 2 μm. **k**, **m** Zoomed-in FIB-SEM image of ROI 1 and 2, respectively, showing a single slice. The arrowheads indicate subsynaptic vesicles. Scale bars for **k** and **m** equal 0.5 μm. Corresponding FIB-SEM stacks are in Supplementary Movies 12 and 13, respectively. **l**, **n** Segmented FIB-SEM volumes corresponding to ROI 1 and 2, respectively; see Supplementary Movies 14 and 15 ($n = 3$ cells for 2 min; 3 cells for 5 min, two independent experiments). See also Supplementary Figure 3

at the activated surface devoid of VAMP7 signal were also observed (Fig. 4i, ROI 2). The corresponding FIB-SEM images revealed an abundance of vesicles at the activated surface as well as in perinuclear regions higher up in the cell (Fig. 4j). The zoomed images show several vesicles within ROIs 1 and 2 near the activation surface (Fig. 4k–n, FIB-SEM slices Supplementary Movie 12 corresponds to ROI 1 and Supplementary Movie 13 corresponds to ROI 2; and segmented volumes are in Supplementary Movies 14 and 15, segmented volume with membranes shown in Supplementary Figure 3d and e). En face (*xy*) views of the entire activated T-cell surface at early and late time points show little evidence of vesicles at the activated cell surface at early

time points, but an abundance of vesicles recruited to the immunological synapse at late time points (Supplementary Figure 3g–j). Thus, the FIB-SEM data corroborate at high resolution and without selective labeling of cellular compartments, the live cell imaging data described above show that microcluster formation at the initiation of T-cell activation occurs independent of vesicles, followed by subsequent recruitment of vesicles to the immune synapse.

**VAMP7+ vesicles track between microclusters on microtubules**. Once we had a complete view of the 3D volume of the

cell during T-cell activation, we examined the activated T-cell surface at later times once the cell was spread on the coverslip and visualized the dynamics of VAMP7-positive vesicles and microclusters with higher spatial and temporal resolution using TIRF-SIM. As ZAP-70 is a stable microcluster marker and represents sites of TCR signaling[3], we imaged cells expressing emerald-VAMP7 and ZAP-70-apple. It became immediately apparent that

VAMP7-positive vesicles preferred to localize to areas that contained ZAP-70 microclusters (Fig. 5a; Supplementary Movie 16). Upon closer inspection, we observed directed VAMP7 vesicle movement between ZAP-70 microclusters. We observed that vesicles (indicated as a gray sphere) appeared to slow down as they approached a microcluster and displayed decreased mobility on the microcluster (Fig. 5b; Supplementary Movie 17). VAMP7

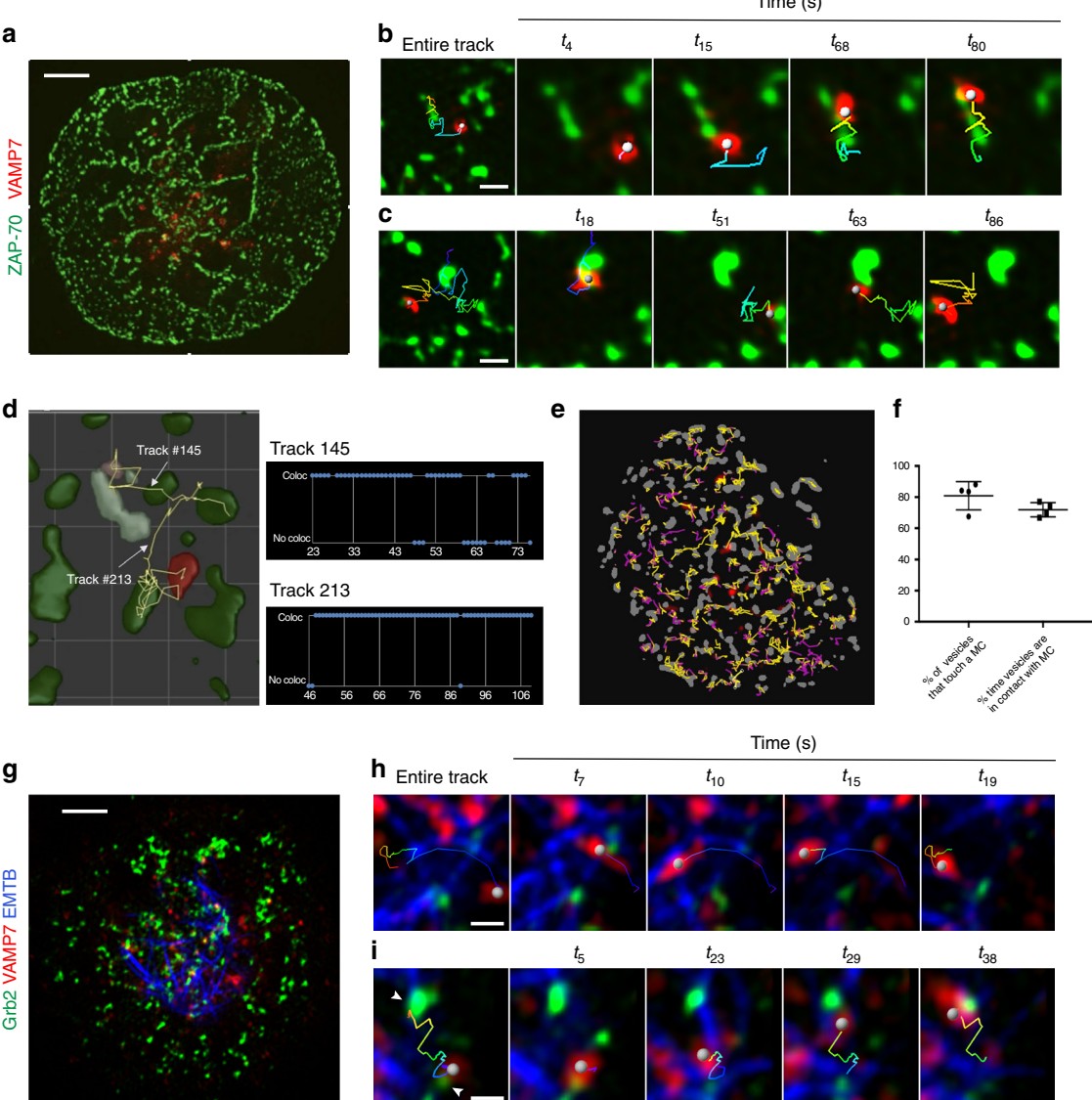

**Fig. 5** VAMP7 vesicles track between microclusters on microtubules at later times after activation. Jurkat cells transfected with indicated constructs were dropped onto stimulatory coverslips and imaged by TIRF-SIM 5 min after initial stimulation. **a–e** Jurkat cells were transfected with ZAP-70-apple and emerald-VAMP7. **a** Image of the entire cell. Scale bar equals 2 μm. **b**, **c** Zoomed-in images focusing on a single VAMP7 vesicle and a few ZAP-70 microclusters. **b** shows a vesicle tracking on a microcluster; **c** shows a vesicle tracking between multiple microclusters (*n* = 7 cells, three independent experiments). Scale bars for b and c equal 0.2 μm. **d** Zoomed-in area of a region of interest of a cell in which vesicles (red) and microclusters (green) have been segmented and show two tracks of VAMP7 vesicles moving between ZAP70 clusters. To the right are time plots of the tracks shown in **d** indicating when colocalization (coloc) or no colocalization (no coloc) between VAMP7 and ZAP-70 was observed. **e** Spatiotemporal map of an entire cell with ZAP-70 microclusters in gray and VAMP7 tracks in yellow (when they colocalized with ZAP-70) or purple (when no colocalization was observed). **f** Graph of the percentage of vesicles per cell associated with microclusters and the frames that a vesicle was associated with a microcluster in the lifetime of its appearance in the TIRF field. **g** Jurkat cells were transfected with EMTB-GFP, Grb2-apple, and Halo-VAMP7. Image of the entire cell. Scale bar equals 2 μm. **h**, **i** Zoomed-in image focusing on a single tracked VAMP7 vesicle. **h** shows a vesicle tracking on a microtubule; **i** shows a vesicle tracking on a microtubule between two Grb2 microclusters indicated by indented arrowheads (*n* = 5 cells, three independent experiments). Scale bars for h and i equal 0.2 μm. In **b**, **c**, **h**, and **i**, the tracked vesicle is indicated as a gray sphere. The entire track is shown and color-coded to indicate time (earliest time point in blue and the latest time point in red). In the first panel, the entire track is shown. The remaining panels show four time points, each with a track that displays the previous five time points in **b** and **c** or previous 20 time points in **h** and **i**

vesicles also appeared to move along paths between multiple ZAP-70 microclusters (Fig. 5c; Supplementary Movie 18). Because ZAP-70 microclusters are stable, we took an image of ZAP-70 microclusters at the beginning of the collection and then imaged VAMP7 vesicle movement at 200-ms intervals. At this faster collection speed, the directed movement of VAMP7 vesicles between ZAP-70 microclusters was even more obvious (Supplementary Movie 19). To quantify the association between vesicles and microclusters in an unbiased way, we segmented vesicles and microclusters and tracked them over the entire time series. We then analyzed the number of frames that a vesicle touched a microcluster in the lifetime of its appearance in the TIRF field. For example, VAMP7 vesicle tracks 145 and 213 were colocalized with a ZAP-70 microcluster for 75 and 95% of their lifetimes, respectively (Fig. 5d; Supplementary Movie 20). When this analysis was extended to all the vesicles that appeared in the TIRF field of the whole cell over the complete time-course, the same trend persisted. A visual representation of this analysis is shown in Fig. 5e, where individual vesicles were tracked over time and their path color-coded according to whether the vesicle touched (yellow) or did not touch (purple) a microcluster. Such dynamic colocalization analysis between vesicles and clusters revealed that 80% of VAMP7-positive vesicles associated with microclusters for 75% of their lifetime in the TIRF field (Fig. 5f), well above randomized association as evaluated by Costes' randomization (Methods).

We then investigated the nature of the cytoskeletal tracks by which the dynamic VAMP7-containing vesicles move between microclusters. VAMP7 vesicles have been shown to move on microtubules from the cell soma to the periphery[13, 14, 20]. We imaged cells co-expressing EMTB-GFP which marks microtubules, Halo-VAMP7, and Grb2-apple as a marker of active microclusters. Microtubules were close enough to the plasma membrane to be within the evanescent field and visualized by TIRF (Fig. 5g; Supplementary Movie 21). In three-color TIRF-SIM movies, VAMP7-positive vesicles displayed curvilinear tracks that were spatially coincident with EMTB-GFP-labeled microtubules. VAMP7 vesicles switched between spherical and tubular morphology as they tracked on microtubules (Fig. 5h; Supplementary Movie 22), and vesicles frequently switched between different microtubule tracks. Furthermore, VAMP7 vesicles tracked on microtubules between Grb2-positive microclusters (Fig. 5i; Supplementary Movie 23), corroborating that VAMP7 vesicle movement is linked to microcluster architecture. To further confirm that VAMP7-positive vesicles moved on microtubule tracks, we destabilized the microtubule network using nocodazole. As in Fig. 1d, a concentration of nocodazole was chosen to not inhibit TCR activation, and the disruption of microtubules by nocodazole was confirmed by visual inspection of microtubules. Nocodazole resulted in a virtually complete arrest of the movement of the VAMP7 vesicles and a strong reduction in the total number of VAMP7 structures in the TIRF field (Supplementary Movie 24), confirming that VAMP7 vesicles interact with and move along microtubules.

**VAMP7$^+$ vesicles are associated with LAT flares at the synapse.** The striking directed movement of vesicles between microclusters and increased dwell time at microclusters led us to question the role of vesicles, after they were recruited to the synapse. We again used LLSM because, due to minimal photodamage afforded by this technique, we could evaluate vesicle and plasma membrane dynamics over the entire cell surface and image later events in T-cell activation. This technique revealed new levels of detail and heterogeneous behaviors in the three-dimensional motion of

vesicles. Individual vesicles also displayed heterogeneity within their trajectories, such as changes in speed and direction. Throughout the cell and at all times, vesicles continually exhibited repeated fusion (merging of two vesicles) and fission (splitting of vesicles) events.

To understand the role of VAMP7-positive vesicles at later times in T-cell activation, we focused on vesicle behavior at the synaptic interface (for examples of vesicle behaviors away from the synapse, see Supplementary Figure 4 and Supplementary Movies 25 and 26). Several phenomena were observed when examining individual vesicles. Due to high temporal resolution, we could track vesicle movement from within the cell to the activated plasma membrane at the synapse (Fig. 6a; Supplementary Movie 27). Interestingly, when a VAMP7-positive vesicle (indicated as a gray spot) approached and made contact with the activated T-cell surface, enhanced LAT fluorescence signal appeared as "flares" at the synapse (Fig. 6a right panel, $t_{152}$, $t_{167}$ and $t_{180}$). We could also track the movement of VAMP7-positive vesicles from the center of the synapse to the periphery (Fig. 6b; Supplementary Movie 28). Again, flares of LAT appeared at the synapse coincident with a VAMP7-labeled vesicle touching down at the activated T-cell plasma membrane (Fig. 6b right panel, $t_{28}$, $t_{32}$, and $t_{35}$). LAT fluorescence separated from VAMP7, as the vesicle touched down at the immune synapse and striking enrichment of LAT at the tip of the VAMP7-positive vesicle was observed (Fig. 6c). It is important to note that the resolution is not sufficient to differentiate whether LAT at the flare is localized at the plasma membrane or on a vesicle. Quantification of LAT and VAMP7 signals within a segmented VAMP7-positive vesicle during a "flare event" showed that both VAMP7 and LAT fluorescence increased at the time of the flare (Fig. 6d). Careful frame-by-frame inspection as a vesicle was touching down revealed that two VAMP7-positive vesicles fused together to cause the increase in VAMP7 fluorescence at the site of the flare (Fig. 6b right panel, $t_{25}$ indented white arrowheads in the top panel point to two VAMP7 vesicles that fuse). LAT fluorescence originating from both the approaching vesicle and the neighboring plasma membrane appears to contribute to the LAT "flare" (Fig. 6b right panel, $t_{28}$, $t_{32}$, and $t_{35}$, white arrowheads in the bottom panel indicate lifting of plasma membrane). Furthermore, after the flare, a decrease in both LAT and VAMP7 signal from the VAMP7-positive vesicle was observed compared to when the vesicle approached the synapse (Fig. 6d compare $t_{38}$ to $t_{19}$), indicating a loss of both LAT and VAMP7 from the vesicle. VAMP7 vesicles at the site of the LAT flares observed at the activated T-cell plasma membrane appeared to be "trapped" and had reduced mobility for several seconds (Fig. 6e; Supplementary Movies 27 and 28). En face views also showed that LAT flares at the immune synapse appear and disappear coincidentally with VAMP7-positive vesicles (Fig. 6f–i). Thus, in contrast to the early time points where we observed that LAT microclusters form independent of vesicle interaction, at later times, a second wave of LAT localized on intracellular VAMP7-positive vesicles is recruited to the synapse. LAT on VAMP7 vesicles is transiently concentrated on vesicles or delivered to the synapse and also appears to cause co-clustering of LAT residing at the plasma membrane. We observe a complex, dynamic, heterogeneous interplay between vesicular and plasma membrane pools of LAT that has not previously been appreciated using conventional light microscopy techniques.

**Microcluster composition before and after VAMP7 recruitment.** We observed LAT microcluster formation before VAMP7 vesicle recruitment, and persistence of LAT microclusters, once vesicles were recruited to the synapse. Therefore, we wanted to

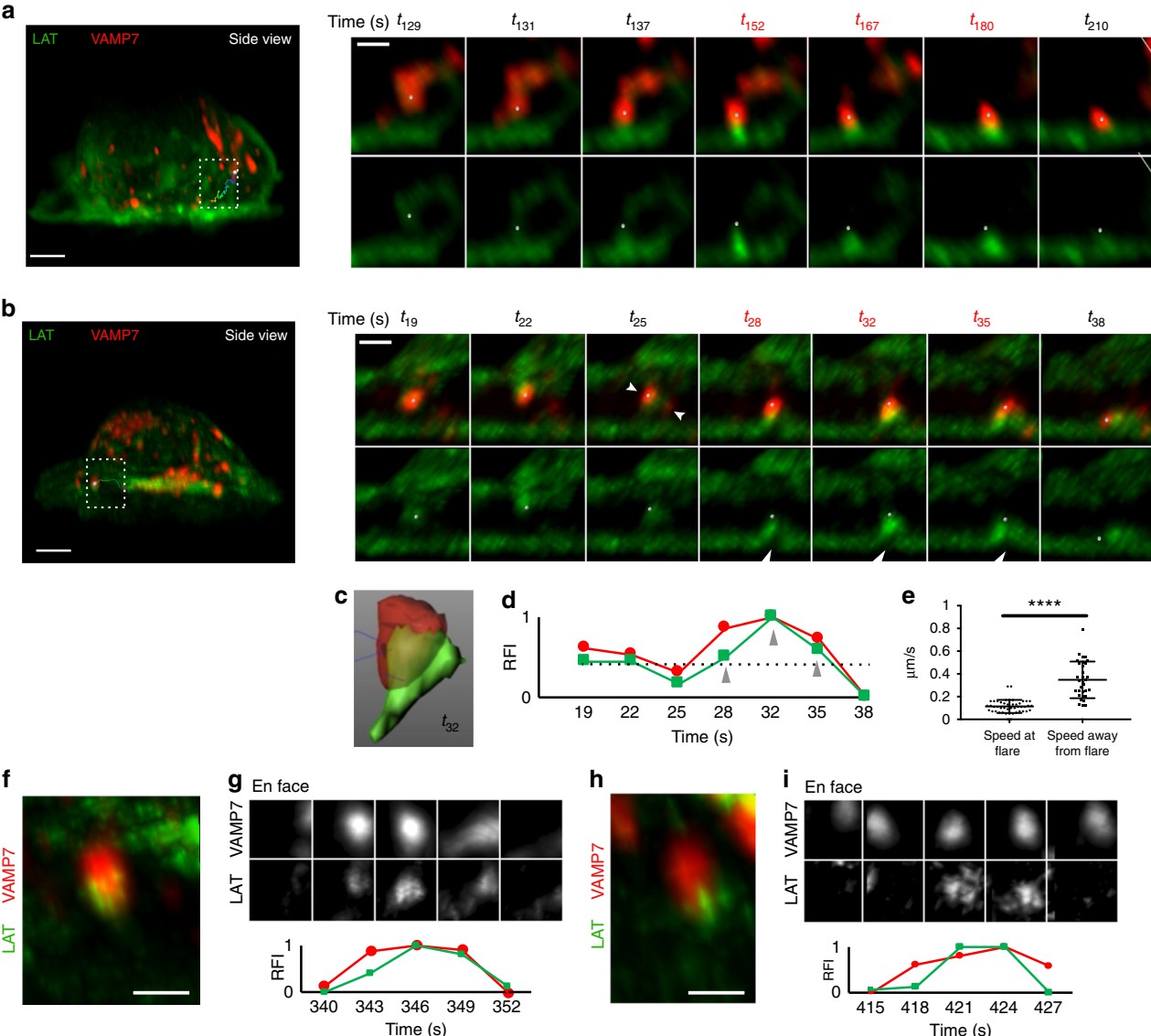

**Fig. 6** LAT and VAMP7 dynamics at later times after activation. Lattice light sheet images of Jurkat cells transfected with LAT-neon green and Halo-VAMP7, dropped onto stimulatory coverslips, and imaged 5 min after initial stimulation. **a**, **b** The left panel shows the side view of the entire cell. The boxed region highlights a vesicle track. The tracked vesicle is a gray sphere. The entire track is shown and color-coded to indicate time, with the earliest time point in blue and the latest time point in red. Right panels show zoomed-in views of the vesicle. The top panel shows LAT (green) and VAMP7 (red), while the bottom panel shows LAT only. **a** A VAMP7 vesicle moving from the inside of the cell to the stimulated surface is shown and in **b** a VAMP7 vesicle moving from the center of the cell to the periphery is shown. In **b** white indented arrowheads in $t_{25}$ top panel indicate two VAMP7 vesicles that fuse. White arrowheads in $t_{28}$, $t_{32}$, and $t_{35}$ indicate the plasma membrane lifting up. Scale bars for **a** and **b** left panels equal 2 μm and right panels equal 0.5 μm. **c** Segmented VAMP7 (red) and LAT (green) at a flare event corresponding to $t_{32}$ in **b**. **d** Relative fluorescence intensity (RFI) plots of VAMP7 (red) and LAT (green) intensity sums within the segmented VAMP7 vesicle shown in **c** over time. Gray arrowheads indicate the times that show the "flare" event. **e** Speed of VAMP7 vesicles away from and at the site of a flare were graphed. **f**, **h** Zoomed-in views of VAMP7 vesicles showing an increase in LAT fluorescence at the vesicle tip as it touches down on the plasma membrane. Scale bars equal 0.5 μm. **g**, **i** Top panels show time series showing LAT and VAMP7 increases in an en face view of the plasma membrane. Lower panels show relative fluorescence intensity (RFI) plots of VAMP7 (red) and LAT (green) ($n = 5$ cells, three independent experiments). See also Supplementary Figure 4

analyze whether LAT microcluster composition changed upon vesicle recruitment. LAT has several tyrosines (Y) that are phosphorylated upon TCR triggering. These phosphotyrosine motifs serve as binding sites for SH2 domain-containing proteins, including PLCγ1, Grb2, and Gads[2]. Gads recruits the adapter SLP-76 to LAT. We decided to simultaneously image LAT and VAMP7 in combination with Grb2, PLCγ1, or SLP-76, to follow three major signaling molecules emanating from the LAT signaling hub.

To analyze the presence of these proteins within LAT microclusters before and after VAMP7 vesicle recruitment, we imaged cells co-expressing LAT, VAMP7, and either Grb2, PLCγ1, or SLP-76 using TIRF microscopy. Cells were imaged from the earliest observable time point before recruitment of VAMP7 vesicles (i.e., no vesicles were visible in the TIRF field), until 6 min of activation when cells were well-spread and robust recruitment of VAMP7 vesicles to the synapse was observed. As described in Fig. 2, we observed the formation of LAT

microclusters within the first few seconds of contact with the stimulatory coverslip, before recruitment of VAMP7 vesicles. LAT microclusters contained Grb2, PLCγ1, and SLP-76 from the earliest observable time point, prior to VAMP7 recruitment

(Fig. 7a–c, $t_0$ left panels). Line scans through microclusters at these early time points show the absence of VAMP7 signal above background, and coincident peak fluorescence intensities of LAT and the associated effector protein, thus confirming colocalization

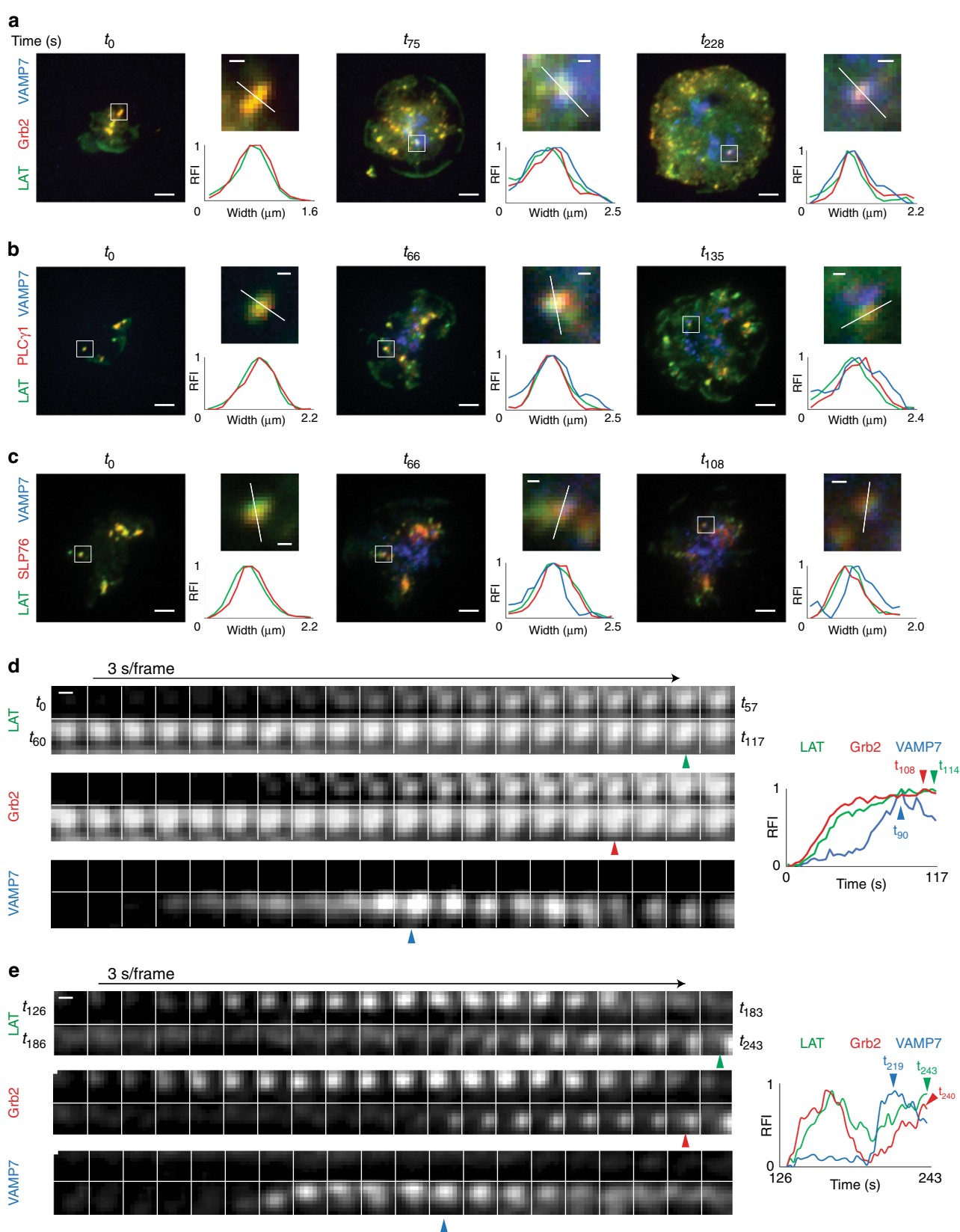

of microcluster components prior to vesicle recruitment (Fig. 7a–c large panels, top-right panels show magnified regions of interest and bottom-right graphs show relative fluorescence intensities across line scans).

Over time, as the cells spread, new LAT microclusters formed as additional areas of the cell contacted the coverslip. VAMP7 vesicles were recruited to the synapse at these later time points and Grb2, PLCγ1, and SLP-76 were detected in LAT microclusters when VAMP7 vesicles were present (Fig. 7a–c, middle and right panels). Line scans through microclusters at these later time points show coincident peak fluorescence intensities of VAMP7, LAT, and the associated effector protein, demonstrating the continued colocalization of microcluster components upon vesicle recruitment. These results indicate that microcluster composition does not change significantly during vesicle recruitment to the immune synapse.

We then analyzed individual microclusters containing LAT, Grb2, and VAMP7 kinetically (Fig. 7d, e). To evaluate recruitment patterns of the molecules, we focused on microclusters that formed during imaging. In Fig. 7d, we followed a region of interest from the earliest observable time point ($t_0$) and observed that LAT and Grb2 signals increased coincidentally as the microcluster formed, indicating that both LAT and Grb2 were recruited simultaneously to the microcluster. We observed recruitment of a VAMP7 vesicle to the microcluster about 50 s after detection of LAT and Grb2 (Fig. 7d, blue arrowhead indicates peak VAMP7 fluorescence at $t_{90}$). After VAMP7 vesicle recruitment to the microcluster, a slight increase in both LAT and Grb2 signals was detected at the microcluster (Fig. 7d, red arrowhead indicates peak Grb2 fluorescence at $t_{108}$ and green arrowhead indicates peak LAT fluorescence at $t_{114}$), even as the VAMP7 signal decreased. A second example is shown in Fig. 7e, where we followed a region of interest in which a microcluster formed at $t_{126}$ after initiation of imaging. Consistent with the region of interest shown in Fig. 7d, simultaneous recruitment of LAT and Grb2 to this microcluster was also observed. Kinetic analysis of LAT and Grb2 signals showed a decrease in fluorescence of both molecules before a VAMP7 vesicle was recruited to the region of interest (Fig. 7e, blue arrowhead indicates peak VAMP fluorescence at $t_{219}$). Strikingly, coincident with the appearance of the VAMP7 vesicle, both LAT and Grb2 signals showed an increase (Fig. 7e, red arrowhead indicates increased Grb2 fluorescence at $t_{240}$ and green arrowhead indicates increased LAT fluorescence at $t_{243}$). Similar trends were observed in microclusters positive for PLCγ1 and SLP-76, where recruitment of a VAMP7 vesicle was coincident with a detectable increase in fluorescence of LAT and the associated effector molecule (Supplementary Figure 5). Thus, we observe oscillations in fluorescence of microcluster components, with increases in fluorescence coincident with vesicle recruitment. These results indicate that vesicles may play a role to maintain sustained signaling from microclusters.

## Discussion

The relevance of LAT localization (plasma membrane vs. vesicles) for TCR signal transduction is a subject of intense discussion. In this study, we used high-resolution, time-lapse multicolor imaging as well as high-resolution fixed cell 3D correlative light and electron microscopy to investigate the key sequence of events by which cell surface and vesicular pools of LAT establish and get recruited to signaling microclusters. Using techniques that image the entire cell volume, we discovered that cell surface LAT laterally translocates to form microclusters and subsequently LAT vesicles decorated with VAMP7 are recruited to the IS. Using correlative light and 3D electron microscopy, we corroborated the live cell imaging results that early microclusters are not associated with vesicles, while later after activation several vesicles are recruited to the IS. Using high-resolution TIRF-SIM microscopy to visualize the IS, we observed that once recruited to the activating surface, VAMP7-positive vesicles moved in a directed manner between microclusters on microtubules. Thus, our data offer spatiotemporal evidence of two phases of early T-cell activation: a rapid phase where proteins at the cell surface are recruited to form microclusters and a subsequent phase, during which vesicles are recruited to the synapse, travel on microtubules, and dynamically interact with microclusters (Fig. 8).

Several controversies surround the functional differences between the two cellular pools of LAT. Questions have arisen as to whether plasma membrane-resident LAT plays any role in LAT activation[12], the timing of vesicular LAT recruitment to the synapse, whether vesicular LAT once recruited to the immune synapse, fuses with the plasma membrane[13, 14], and whether LAT phosphorylation occurs at the plasma membrane or in vesicles[11, 12]. Our study resolves some of these controversies and makes a number of new observations. Previous studies that proposed vesicular recruitment of LAT as the predominant mechanism by which T cells get activated, focused on cells that had been activated for 5–10 min[11, 12, 14], whereas peak LAT phosphorylation occurs between 1 and 2 min after TCR engagement[26]. To clarify these discrepancies, we focused our imaging studies to include time points from the instant a T cell became activated until 10 min after activation. We saw that upon stimulation, the first phase of LAT microclusters aggregates rapidly, in seconds, from redistribution of cell surface LAT even when intracellular vesicles were several microns away from the synapse. Thus, it appears that one reason for contrasting results among the various studies is the differences in activation times used by the various groups. Another likely reason is studies that proposed that microclusters form by exocytosis of vesicular LAT silenced fusion proteins VAMP7 and Syt7[13, 14]. Knockdown of these proteins over a period of 24 h could affect a number of trafficking events prior to T-cell activation, thus abrogating microcluster formation in an indirect manner. In contrast, when we acutely inhibited calcium-mediated exocytosis or vesicle recruitment using inhibitors for a few minutes prior to activation, neither process was required for the first phase of LAT microcluster

**Fig. 7** Microcluster composition before and after VAMP7 recruitment. **a–c** Jurkat cells were transfected with **a** LAT-Halo, Grb2-scarlet, and emerald-VAMP7, **b** LAT-ruby, PLCγ1-Halo, and emerald-VAMP7, or **c** LAT-ruby, SLP76-Halo, and emerald-VAMP7, dropped onto stimulatory coverslips, and imaged by TIRF microscopy. Indicated time points are shown with $t_0$ corresponding to the earliest observable time point. For each time point, the upper-right image was magnified from the region marked by a white box in the left image, and the bottom-right graph shows the relative fluorescence intensity (RFI) measured across the width of the white line in the corresponding upper-right image. **a–c** Scale bars in left images equal 2 μm. Scale bars in upper-right images equal 0.5 μm ($n = 6$ cells for Grb2, 4 cells for PLCγ1 and SLP-76; three independent experiments). **d**, **e** Left panels show time-lapse montage at 3 s/frame graphs of a single microcluster and kinetics of recruitment of LAT, Grb2, and VAMP7. On the right, relative fluorescence intensity (RFI) plots of the time-lapse montage are shown. Blue arrowheads indicate peak VAMP7 fluorescence, red arrowheads indicate peak Grb2 fluorescence (**d**) or increased Grb2 fluorescence once VAMP7 is recruited (**e**) and green arrowheads indicate peak LAT fluorescence (**d**) or increased LAT fluorescence post VAMP7 recruitment (**e**). Scale bars equal 0.5 μm. See also Supplementary Figure 5

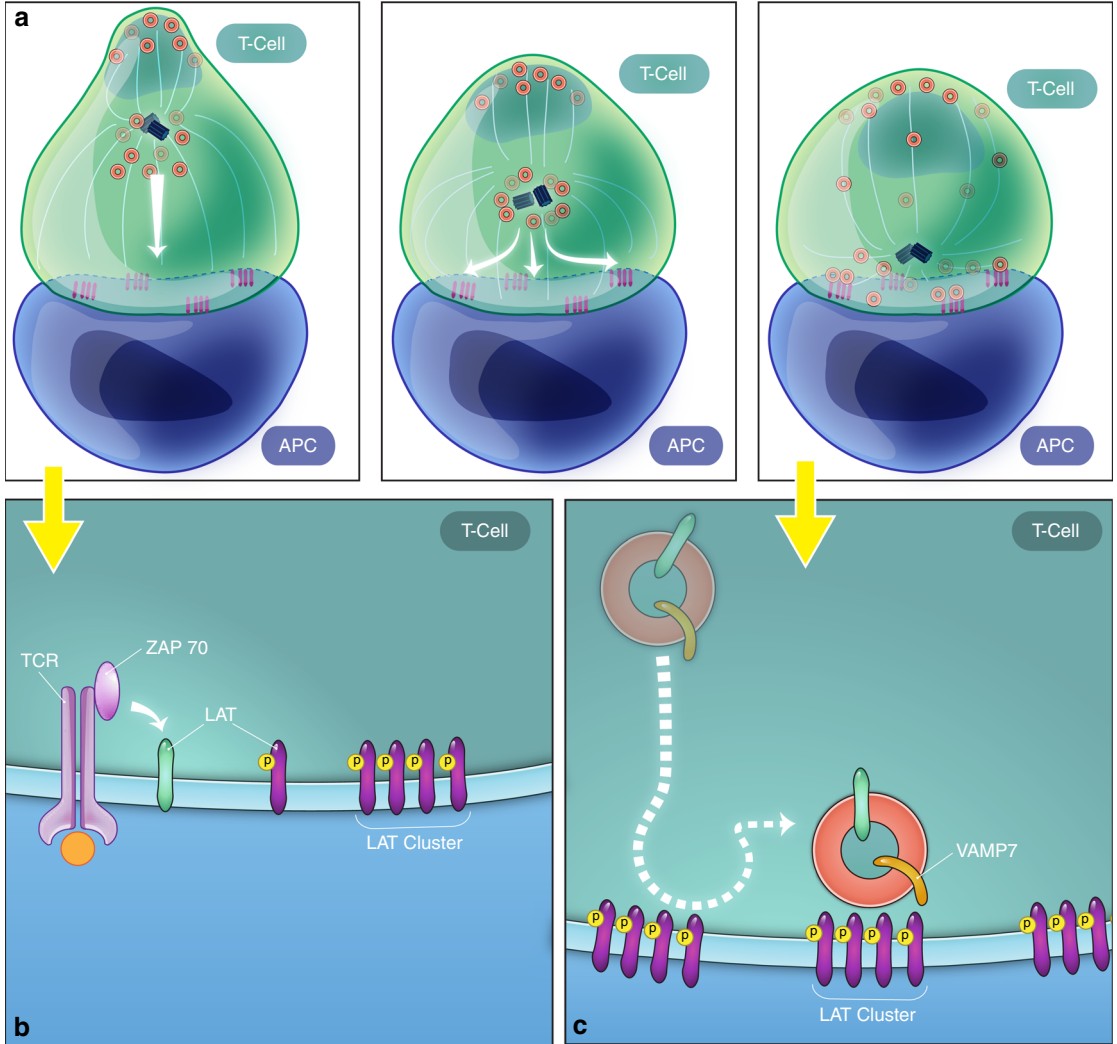

**Fig. 8** Two-phase model for LAT activation. A model for LAT microcluster formation and maintenance using the two cellular pools of LAT is shown. **a** The left panel shows that at early time points, LAT vesicles decorated with VAMP7 are several microns away from the immune synapse, while plasma membrane-resident LAT can move laterally to be recruited to microclusters. The middle and right panels show that microtubule polarization toward the immunological synapse causes the reorientation of various vesicular compartments to the subsynaptic zone. **b**, **c** Zoomed-in regions of early and late stages of activation, respectively. **b** At early time points, plasma membrane LAT is recruited to microclusters and activated ZAP-70 phosphorylates LAT at the PM. **c** At later time points, LAT/VAMP7 vesicles are recruited to the immune synapse, move in a directed manner between microclusters, and interact dynamically with microclusters

formation or phosphorylation, arguing against a major role for vesicle recruitment or exocytosis in microcluster formation.

We observed that in the second phase, vesicles containing VAMP7 and LAT are recruited to the stimulated surface coordinated with MTOC localization to the synapse. Once VAMP7-positive vesicles were recruited to the IS, directed movement of vesicles on microtubules between microclusters was observed. Not only did vesicles move between clusters, but they also displayed decreased mobility and increased dwell times at clusters. Thus, VAMP7 vesicle movement appears to be defined by microcluster architecture similar to what has been observed previously with lipophilic tracers[11]. The decreased mobility of vesicles at clusters is consistent with high-resolution imaging studies that described that LAT signaling clusters may represent discrete subdomains in T cells[27] and previously described plasma membrane confinement zones[28]. The directed movement of vesicles raises the question of what feature of clusters specifically attracts and retains vesicles? Evidence suggests that t-SNARE

subcomplexes localize to target membranes to serve as guides for v-SNAREs on vesicles to bind and facilitate fusion[29]. t-SNAREs could localize at microclusters to allow assembly of the SNARE complex and specific fusion at microclusters. Indeed, t-SNAREs SNAP-23, and syntaxin-4 cluster at the IS and play an important role in the targeting of TCR vesicles to the IS[30]. Thus, a critical next step will be to investigate the localization of t-SNAREs and other proteins involved in vesicle docking and fusion and evaluate whether they form active zones at microclusters. In several biological systems, vesicle docking and exocytosis is spatially confined to distinct regions of the plasma membrane[31] and it will be important to investigate whether microclusters are "targeting patches" or "hotspots" that control localized docking and/or exocytosis.

The striking directed movement of VAMP7-positive vesicles between microclusters once they are recruited to the immune synapse, also raises the question of their role at microclusters. Previous studies have indicated that vesicle recruitment to IS

plays a role in sustained signaling[32–34] and vesicle proximity to microclusters has been associated with LAT phosphorylation[11]. As VAMP7 is involved in the exocytosis of cargos in many systems[35–39], and VAMP7-positive vesicles contain LAT, an expected role of VAMP7 is LAT delivery to the IS for microcluster maintenance and sustained signaling. In this regard, we observed by TIRF microscopy oscillations in microcluster fluorescence and an increase in LAT as well as LAT-interacting proteins upon vesicle recruitment to the synapse. Moreover, striking increases in LAT intensity at the IS where VAMP7-positive vesicles touched down were captured on the lattice light sheet microscope (LLSM). As LAT is present in the VAMP7-positive vesicles before they touch down at the synapse, these "flares" could represent delivery of LAT molecules to the synaptic plasma membrane in a fusion process. Since we did not observe a complete collapse of VAMP7-positive vesicles at the plasma membrane as would be expected for total vesicular fusion, these events may represent "kiss-and-run" exocytosis in which a vesicle opens and closes transiently, retaining its gross shape[40]. Alternatively, the LAT "flare" could represent a concentration of LAT while still residing on docked vesicles without fusion, as has been previously proposed[14]. Notably, LAT originating from the plasma membrane adjacent to the site of vesicle contact also appeared to contribute to the LAT "flare" raising the possibility that vesicular LAT co-clusters in trans with plasma membrane-resident LAT, consistent with LAT oligomerization that has been observed both in vitro and in vivo[41, 42]. In the opposite direction, given that this v-SNARE is involved in late-endosome to lysosome transport[43–46], VAMP7-positive vesicles could be involved in endocytic trafficking at microcluster sites. We cannot distinguish between these possibilities by imaging VAMP7 and LAT alone. A better understanding of these results requires visualizing these events simultaneously with other labeled components that coordinate vesicle traffic such as Rab GTPases and various endocytic and exocytic markers.

Vesicular traffic at the IS has emerged recently as an important component in T-cell activation[47–50]. Recent studies have demonstrated that coordinated movement of exocytic vesicles containing signaling molecules toward the immune synapse is necessary for TCR signaling[13, 14, 51]. Vesicular traffic is also involved in sustaining TCR activation[30], signal termination[52], directed cytokine secretion[53], and the transfer of exosomes or microvesicles from the T cell to the APC[54, 55]. Concurrently, signals at the plasma membrane initiate endocytosis and the formation of endosomes in which specialized signals may be propagated to different cellular locations[56]. In conclusion, we have made use of recent advances in imaging techniques to gain insights into the dynamic interplay between plasma membrane and vesicular pools and how they mediate spatiotemporal control of T-cell signaling. From our observations, we propose a two-phase model of T-cell activation in which plasma membrane-resident proteins are first recruited to form microclusters followed by vesicular recruitment to the synapse and directed vesicle movement between microclusters. A better understanding of the mechanisms, locations, and the precise roles of plasma membrane versus vesicular signals in T-cell activation may shed light on the role that intracellular trafficking plays in normal T-cell function and in T-cell-mediated diseases.

## Methods

**DNA constructs**. The constructs used in this paper were generated as follows: LAT-neon green was generated by cloning LAT cDNA into AgeI-NotI sites of pNeon-Green N1 (gift from Waterman lab, NHLBI, NIH). GFP-VAMP7 was a gift from Gillian Griffiths (Cambridge Institute for Medical Research). To obtain brighter VAMP7 signals, Halo-VAMP7 was generated by cloning VAMP7 cDNA into PvuI-NotI sites of pHTNHalo-tag vector (Promega, catalog no. G7721) and emerald-VAMP7 was generated by cloning AgeI-BsrGI-digested emerald sequence from emerald-zyxin6 (Addgene, plasmid no. 54319) into VAMP7 cDNA sequence. EMTB-GFP was a gift from the Hammer lab (NHLBI, NIH). ZAP-apple and Grb2-apple were generated by cloning in AgeI-BsrGI-digested apple cDNA from mAppleN1 (Addgene, plasmid no. 54567) into AgeI-BsrGI-digested plasmids containing human ZAP-70 and Grb2 cDNA sequences in mYFPN1 plasmid (Clontech). LAT-ruby was generated by cloning LATcDNA into AgeI-NotI digested mRubyN1 plasmid (Addgene 54581). PLC γ1-Halo and SLP76-Halo were generated by cloning PLC γ1 and SLP76 cDNA into NheI-ApaI digested pHTC-Halo-tag vector (Promega, catalog no. G7711). Grb2-scarlet was generated by cloning Grb2 cDNA into NheI-AgeI digested pLifeAct_mscarlet_N1 plasmid (Addgene, plasmid# 85054).

**Reagents**. Human anti-CD3ε (UCHT or HIT3a) monoclonal antibodies were purchased from Pharmingen and were used to coat coverslips for imaging assays and OKT3 (produced in the laboratory) and was used to trigger T-cell activation in calcium flux assayed by flow. The following antibodies were used for immunostaining: mouse anti-pLAT[226] (BD Biosciences, catalog no. 558363) and mouse anti-pY clone 4G10 (Millipore, catalog no. 05-321). BAPTA-AM, indo-1 AM, and nocodazole were from Invitrogen. Halo-tag ligand conjugated to Janelia Fluor (JF)-646 and -549 were gifts of Luke Lavis at Janelia Research Institute.

**Cell culture and transient transfection of Jurkat cells**. Jurkat E6.1 cells have been described previously[57]. All Jurkat cells were cultured in RPMI 1640 supplemented with 10% fetal bovine serum and antibiotics. For transient transfections, $1 \times 10^6$ Jurkats were transfected with 2 μg of DNA with the Nucleofector Kit V (Lonza, catalog no. VCA-1003), Program H-10, or Program X-001 24 h prior to imaging.

**BAPTA and nocodazole inhibition**. To prevent any elevation of intracellular calcium, cells were preloaded with BAPTA-AM (20 μM) for 20 min at 37 °C in complete media buffered with 1 mM EGTA. BAPTA-loaded cells were rinsed and injected into chambers supplemented with 1 mM EGTA. The effectiveness of the treatment was confirmed using Jurkat cells loaded with indo-1AM. For nocodazole inhibition, cells were treated in complete medium with 2 μm nocodazole for 5 min and were injected into chambers supplemented with 2 μM nocodazole.

**Immunofluorescence**. Jurkat T cells were allowed to adhere to the substrate for 2 min at 37 °C and then fixed for 30 min in 4% (wt/vol) PFA solution (Electron Microscopy Sciences) in imaging buffer (1 × RPMI without phenol red, 10% FBS, and 25 mM Hepes). Samples were permeabilized in 0.1% Triton-X-100 for 3 min and then incubated in a blocking solution consisting of 10% FBS (Sigma-Aldrich), 0.01% sodium azide (Sigma-Aldrich), and 1 × PBS for 1 h at room temperature (RT). After three washes in 1 × PBS, the cells were stained with primary antibody in blocking solution (anti-pLAT[226] from BD Biosciences, catalog no. 558363 was used at 20 μg/ml; anti-pY clone 4G10 from Millipore, catalog no. 05-321 was used at 2 μg/ml) for 1 h at RT, followed by secondary antibody in blocking solution (isotype-specific Alexa Fluor-conjugated secondary antibodies were used at 1:1000-fold dilution) for 45 min at RT. Images from fixed cells were collected with a Zeiss 710 LSCM, using a 63×, 1.4 NA objective (Carl Zeiss Inc, Thornton, NY).

**Live cell imaging**. Halo-tagged constructs were labeled with 0.5 nM Janelia Farm Halo-647 ligand for 30 min at 37 °C. Cells were imaged after three washes in complete medium and then resuspended in HBS imaging buffer (20 mM Hepes, pH 7.2, 137 mM NaCl, 5 mM KCl, 0.7 mM Na₂HPO₄, 6 mM D-glucose, 2 mM MgCl₂, 1 mM CaCl₂, and 1% BSA). All chambers used for live cell imaging contained HBS imaging buffer.

TIRF images from live cells were collected with a Nikon Ti-E inverted microscope, using a ×60, 1.49 NA objective (Nikon Instruments Inc, Melville, NY) in TIRF mode. Images were captured with an iXon DU88 EM-CCD camera (Andor).

The LLSM used in these experiments is housed in the Advanced Imaged Center (AIC) at the Howard Hughes Medical Institute Janelia research campus. The system is configured and operated as previously described (Chen, Science, 2014). Samples were illuminated by lattice light sheet using 488-nm or 641-nm diode lasers (MPB Communications) through an excitation objective (Special Optics, 0.65 NA, 3.74-mm WD). Fluorescent emission was collected by detection objective (Nikon, CFI Apo LWD 25XW, 1.1 NA) and detected by a sCMOS camera (Hamamatsu Orca Flash 4.0 v2). Acquired data were deskewed as previously described (Chen, Science, 2014) and deconvolved using an iterative Richardson–Lucy algorithm. Point-spread functions for deconvolution were experimentally measured using 200-nm tetraspeck beads adhered to 5-mm glass coverslips (Invitrogen, Catalog # T7280) for each excitation wavelength. Time-lapse, 3D volumetric data were imported into Imaris (Bitplane, Andor) for rendering and visualization.

For conjugate assays, Raji B cells were pulsed with 2 μg/ml SEE for 1.5 h at 37 °C and washed twice in serum-free medium. Raji cells were then labeled with PKH26 as follows: Raji B cells were resuspended in 250 ul of diluent C. A 1:20 dilution of

PKH26 (made in diluent C) was added at 1:1 ratio and cells were incubated for 1 min. Cells were then pelleted and resuspended in imaging buffer for drops.

TIRF-SIM imaging was performed as described previously[17]. In brief, images were acquired using an inverted microscope (Axio Observer; ZEISS) fitted with a ×100 1.49 NA objective (Olympus) and a spatial light modulator to provide structured illumination.

**Image analysis**. Imaris software (Bitplane, Andor) was used for calculation of cluster intensity and cluster area in fixed cells (Fig. 1a, b, e, f). Intensity-based thresholding was used to generate "surfaces" of microclusters in the various channels and area and intensity values for segmented microclusters were obtained. Statistical significance was determined by two-tailed *t*-test. ***$P \leq 0.001$ and **$P \leq 0.01$.

Imaris software (Bitplane, Andor) was used for calculation of VAMP7 vesicle distances from LAT microclusters from LLSM data presented in Fig. 3c as follows. Vesicles from the red channel and microclusters from the green channel were segmented into "surfaces" based on intensity thresholding. For each frame series, we then inspected the images and iteratively evaluated intensity-threshold values to find the most reliable segmentation of vesicles and microclusters into "surfaces". The red surface was then divided into individual vesicles using the "split into spots" feature. The spots were tracked automatically using the "track surfaces" feature and the tracks were then evaluated iteratively to find the most reliable tracking of vesicles. The "Distance Transformation" tool was then used to map the distance of each VAMP7 "spot" from the LAT microcluster "surface" over time. Graphs were generated in Imaris.

Imaris software (Bitplane, Andor) was used for calculation of VAMP7 vesicle speeds from LLSM data presented in Figure 6Bv as follows. Vesicles from the red channel were segmented into "spots" based on intensity thresholding. The spots were tracked manually using the "track spots" feature for time frames that the spot was at the flare event and for at a minimum of five frames before and after the flare event. The speeds of the VAMP7 "spots" were then exported into Excel for five flare events and time frames were categorized as "flares" and "away from flares" based on evaluation of the tracks. Graphs were generated in Graphpad Prism.

Vision4D (Arivis) was used for tracking of VAMP7 vesicles and colocalization of analysis of VAMP7 vesicles with microclusters from TIRF-SIM data (Fig. 5b). Data were filtered, segmented, and tracked as follows. We first applied a type of image-restoration filter (called 2D particle enhancement in Vision4D) that enhances detection of feature points in frame images (time series) that greatly reduced the detection of questionable positive signals coming from background and enriched for true signals coming from objects (our observation). For each frame series, we then inspected the images and iteratively evaluated intensity-threshold values to find the most reliable segmentation of vesicles and microclusters into objects. A segment generator was used to convert the signals above the threshold into objects. The Vision4D software provides an object-based colocalization operation that tags objects when they meet user-defined criteria. In this case, we set the operation so that when a vesicle touches a microcluster, it is tagged as being colocalized with the microcluster. This operation was performed on all frames of each time series. The movement of the objects over time was then tracked with a segment-tracking operation and all the results transferred to a spreadsheet for quantitation.

Costes' randomization algorithm was performed in Image J to rule out the possibility that the observed colocalization of microclusters and VAMP (TIRF-SIM movies shown in Fig. 5) is attributable simply to random noise. Costes' randomization algorithm generates a number of images populated by various amounts of noise in each color channel and calculates the Pearson's coefficient for each one. Costes' randomization algorithm shows that the Pearson's coefficients are due to signal rather than noise.

Adobe PhotoShop and Illustrator (Adobe Systems Inc, San Jose, CA) were used to prepare composite figures.

Image J was used to measure the distribution of LAT and Grb2, PLC, SLP-76, and VAMP7 at microclusters in Fig. 7a–c by drawing line scans across a microcluster. After background subtraction, the line scans were normalized for intensity. For analysis of fluorescence intensity over time (Fig. 7d, e and Supplementary Figure 5), the Time Series Analyzer plugin in Image J was used. A region of interest was cropped and fluorescence intensity over time for all colors was obtained and normalized for intensity.

**FIB-SEM sample preparation**. Jurkat cells were dropped onto stimulatory coverslips and fixed at 2 min and 5 min with 4% (wt/vol) PFA solution (Electron Microscopy Sciences) in 0.1 M sodium phosphate buffer, pH: 7.4 for 30 min. Cells were labeled with CD45-647 (BD Biosciences) in blocking solution consisting of PFN (PBS + 10% fetal bovine serum + 0.1% sodium azide) and 2% goat serum (Jackson Immunolabs) for 20 min at room temperature to label the plasma membrane. After the cells were rinsed thrice gently with 0.1 M sodium phosphate buffer at pH 7, they were blocked and permeabilized for 1 h in a block-perm solution consisting of 2% goat serum (Jackson Immunolabs), 0.1% sodium azide (Sigma-Aldrich), and 0.1% saponin in PBS. Cells were then incubated in primary antibody in block-perm solution for 1 h at room temperature. After three washes in 1 × PBS, the cells were stained with secondary antibody (1:1000-fold dilution in 1 × PBS) in wash-block solution consisting of 2% goat serum (Jackson Immunolabs), 0.1% sodium azide (Sigma-Aldrich), and 0.05% saponin in PBS. After two washes in PBS, cells were DAPI stained for 3 min. After two more washes in PBS, images were captured using a Leica SP8 laser-scanning confocal microscope using a ×63, 1.4 numerical aperture (NA) objective (Leica Microsystems Inc, Buffalo Grove, IL). After the images were collected, cells were fixed with 2.5% (wt/vol) gluteraldehyde (Electron Microscopy Sciences) in PBS for 30 min. After three washes in PBS, cells were left overnight in PBS. The following morning, cells were washed three times for 5 min each in PBS and one time with 0.2 M cacodylate buffer at pH 7.4 (Electron Microscopy Sciences). Cells were then incubated in 1% osmium tetroxide and 1.5% potassium ferrocyanide in 0.2 M cacodylate buffer at pH 7.4 for 1 h in the dark. After one wash with dd water, cells were dehydrated through graded ethyl alcohol (ETOH) in 10-min incubation steps as follows: 2 × 35% ETOH, 2 × 50% ETOH, 2 × 70% ETOH, 2 × 95% ETOH, and 3 × 100% ETOH. Then cells were embedded in resin as follows: 2:1 mix of ETOH and Embed mix 1 h, 1:1 mix of ETOH and Embed mix 1 h, 1:2 mix of ETOH and Embed mix 1 h, 1:2 mix of ETOH and Embed mix 1 h, and 100% Embed mix overnight. Add 100% fresh Embed mix the next morning and incubate in a 60 °C oven for 48 h. Embed mix is 2 ml of Mix A and 8 ml of Mix B + 200 μl of BDMA (Ted Pella), where Mix A is 30.5 g of Eponate 12 Resin (Ted Pella) + 39.2 g of DDSA (Ted Pella) and Mix B is 30.5 g of Eponate 12 Resin (Ted Pella) and 26.2 g of NMA (Ted Pella). After polymerization, the blocks were immersed in liquid nitrogen to separate the gridded glass coverslips from the resin block. The laser-etched pattern on the grid was transferred to the resin, allowing for subsequent location by SEM imaging. The blocks were then gently cleaned, affixed to an SEM stub with conductive silver paint, and sputter coated with a thin conductive layer of carbon before transfer to the FIB-SEM instrument.

**FIB-SEM imaging**. FIB-SEM imaging was performed in a Zeiss Crossbeam 540 (Carl Zeiss Inc.) in conjunction with ATLAS3D software (Fibics Inc.) as previously published[25], with a few modifications. The altered protocols were critical in minimizing imaging artifacts in the area of the resin proximal to the top surface, which encompassed the synaptic region of the activated cells. Briefly, the thin layer of carbon and gridded pattern allowed for the cells that were now just under the resin surface, to be identified by SEM imaging at 3-kV incident energy. Cells that were previously imaged by LM were then protected by a 1-μm-thick carbon pad deposited by the FIB operated at 300 pA. These layers not only provided an extra protection to the surface-proximal region from FIB milling during imaging, but they also provided m/z contrast between the protective pad and the heavy- metal-stained membranous structures of the immune synapse. A platinum- and carbon-patterned protective pad was then deposited with the FIB operated at 700 pA, and data collection was executed with the FIB and SEM operated simultaneously. The FIB operated at 30 kV, 700 pA and SEM operated at 1.5 kV, 1 nA, and the back-scatter signal was recorded at the in-column EsB detector operated with a 900-V grid voltage. In our hands, larger FIB currents resulted in topological artifacts that altered the 3D reconstructions of membranous structures critical to the experiment. The "ROI" images were acquired at 3- or 4-nm pixel sampling and 9- or 12-nm milling increments, with total dwell time of ~3 μs per pixel, and in some cases, "keyframe" images were acquired at lower resolutions (10 × 10 × 100 nm, ~2 μs of dwell time). An imaging run covering an entire cell typically lasted ~40 h and generated a stack of 1000–2000 high-resolution images. These images were processed using in-house IMOD- (REF: Kramer J Struct Biol 1996) based scripts to yield registered, cropped, binned, and inverted isotropic.mrc volumes. In the absence of correlative fiducial markers, "slabs" of volumetric data corresponding to the immune synapse from the confocal and FIB-SEM volumes were aligned using simple transforms in 3DSlicer (Federov A Magn Reson Imaging 2012, slicer.org). Features of interest from FIB-SEM image volumes were segmented and rendered using 3DSlicer, with appropriately sized and placed spherical markers used to denote enclosed membranous compartments.

**Data availability**. The FIB-SEM imaging data that support the findings of this study are available in the National Cancer Institute Center for Strategic Scientific Initiatives Data Coordinating Center (https://cssi-dcc.nci.nih.gov/cssiportal/view/5ac3e62d37384e051c7ab310/). Other data that support the findings of this study are available within the article and its Supplementary Information files or from the corresponding author upon request.

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

## Acknowledgements

This research was supported by the Intramural Research Program of the NIH, NCI, and CCR. This project has been funded in whole or in part with Federal funds from the National Cancer Institute, National Institutes of Health, under Contract No. HHSN261200800001E. The content of this publication does not necessarily reflect the views or policies of the Department of Health and Human Services, nor does mention of trade names, commercial products, or organizations imply endorsement by the U.S. Government. We thank the Advanced Imaging Center at Janelia Farms for access to the lattice light sheet and TIRF-SIM microscopes. The Advanced Imaging Center is jointly supported by the Howard Hughes Medical Institute and the Gordon and Betty Moore Foundation. We thank the following people for their generosity: Luke Lavis for his

contribution of substantial amounts of Halo-tag ligands, Gillian Griffiths for the GFP-VAMP7 construct, John Hammer III for the EMTB-GFP construct, Valarie Barr, Itoro Akpan, and Kunio Nagashima for technical help, Roberto Weigert, John Hammer III, and Valarie Barr for helpful discussions, and Roberto Weigert and Valarie Barr for critically reading the manuscript.

## Author contributions

L.B., J.Y., T.N., K.M.M., A.S.H., and K.N. performed the experiments; L.B. and L.E.S designed the study; L.B. and T.N. performed image analysis; A.S.H. and K.N. performed FIB-SEM experiments and K.N. did FIB-SEM post processing and registration; and L.B. prepared figures and wrote the manuscript with comments from J.Y. and L.E.S.

## Additional information

**Competing interests:** The authors declare no competing interests.

