## [Peer Review File · Nature Communications]

Reviewers' comments:

Reviewer #1 (Remarks to the Author):

In addition to its localization at the plasma membrane of resting T cells, Lat also resides on intracellular vesicles. Analysis of T cells deficient in VAMP-7, suggested that the docking of Lat-containing intracellular vesicles is mandatory for proper TCR signaling. However, given that Lat molecules localized to the plasma membrane (PM) are efficiently phosphorylated in the seconds that follow engagement of the TCR, it has been argued that vesicular Lat molecules are required to sustain signaling only at late time points. Therefore, although indisputable experimental evidence supports the proposal of the existence of a pool of vesicular Lat molecules, contrasting views remain as to its exact function. The present study aims at solving this controversy by assessing the dynamics of the two existing pools of Lat molecules. Using pharmacological agents (BAPTA/EGFTA and nocodazole), the authors first established that neither vesicle recruitment nor calcium-mediated exocytosis are required for LAT microcluster formation at the PM, suggesting that the latter occurs independently of VAMP7-associated vesicles. To define the relationship between the pool of Lat found at PM and associated with intracellular vesicles and their relative contribution to Lat microcluster formation, they used lattice light sheet microscopy (LLSM) to simultaneously image Lat and a vesicular marker VAMP7-positive vesicles. They demonstrated that there exists a lag of 45-60 s between LAT microcluster formation at the PM and recruitment of the VAMP7-positive vesicles to the synapse. A finding further supported by Total Internal Reflection Fluorescence Simulated Interference Microscopy using Lat, Grb2, Zap70 and VAMP-7 reporters and by correlative 3D light and Focused Ion Beam Scanning Electron Microscopy. Although this study does not provide a final answer concerning the role played by the Lat-containing intracellular vesicles that dock at the immunological synapse 45-60 seconds after initiation of activation, the present study that capitalizes on the most advanced imaging tools provides a novel and very precise view on the dynamics of the two pools of Lat. Moreover, the dynamics relationships that exist between the VAMP7-positive vesicles and the PM areas that contained ZAP-70 microclusters, the coincidence of VAMP7 vesicle movement and microtubules, and the enhanced LAT fluorescence that occurs when a VAMP7-positive vesicle approached and made contact with the activated T cell surface are particularly fascinating. The discussion fairly analyzes these novel and exciting results.

Specific (minor) comments

1/ In Fig. 4Ai, ROI 2, the spotted VAMP7-positive vesicle located further above in the cell volume appears associated with pLAT. This is bit puzzling since no pLAT should be present so far away from the stimulatory coverslip. The authors should comment on that issue.

2/ It will be interesting to know the extent of T cell activation that occurs during the first 30-60 s of activation prior to relocalization of the pool of vesicular LAT at the IS. For instance, using the Jurkat model at what time point is it possible to monitor the onset of ERK phosphorylation? Along the same line is ERK phosphorylated after nocodazole treatment?

3/ If I understand properly the reasoning of the authors, they suggest that in the experiments reported by Hivroz and colleagues, deletion of VAMP7 is 'chronic' and might adventitiously affect the constitution of the pool of Lat found at the PM. Since VAMP7 is of much interest for neurobiologists, is there any drugs of optogenetics devices permitting an 'acute' ablation of VAMP7?

Reviewer #2 (Remarks to the Author):

The group recently published a study in JI that makes a strong case that plasma membrane LAT clusters transiently with TCR, is phosphorylated at TCR microclusters and makes up 90% of phosphorylated LAT measured at 2 minutes. This seemed to leave a small potential role for phosphorylation of LAT that is intracellular at the onset of activation, making up approximately 40% of cellular LAT. This study uses state of the art 3D and eTIRF-SIM imaging approaches to further evaluate the role of sub-synaptic vesicles containing LAT in initiation of TCR signaling. They

also include an analysis of VAMP-7 containing vesicles, which carry a fraction of the intracellular LAT and could deliver it to the plasma membrane in a Ca²⁺ dependent manner. The study supports the conclusion of the earlier work in JI that plasma membrane LAT is the main pool used for early signaling. The first two experiments in the paper, one involving Calcium flux buffering and another involving microtubule depolymerization, both immediately point to a problem with control of TCR microcluster growth or signal termination. The CLEM analysis further demonstrates that vesicular compartments near the synapse include many tubulovesicular structures with intraluminal vesicles, which are known as multivesicular bodies and are involved in endosome to lysosome trafficking of ubiquitinated membrane proteins. In fact, LAT is known to be ubiquitinated and when these lysine are changed to arginines TCR signaling is positively dysregulated (Kunii N, Zhao Y, Jiang S, Liu X, Scholler J, Balagopalan L, Samelson LE, Milone MC, June CH. Enhanced function of redirected human T cells expressing linker for activation of T-cells that is resistant to ubiquitylation. *Hum Gene Ther*. 2012. Epub 2012/09/25. doi: 10.1089/hum.2012.130. PubMed PMID: 22998346.). The authors may already recognize these signatures as the discussion clearly cites some relevant papers regarding endocytosis, signal termination through ESCRT function, etc, but the descriptive components of the paper fail to call attention to these features. The rest state an earlier hypothesis first proposed by Purbhoo et al and Soares et al, that the vesicular LAT delivery is important for sustained signaling. My feeling is that the current paper generates no new conclusions, but at the same time is full of missed opportunities to recognize its own novel results and design new experiments that test the novel initial results. At present it would have a low priority based on the lack of any conceptual novelty is written, but a fantastic potential if it were to put together the obvious new thread the authors have uncovered some targeted experiments to test the new hypothesis that the first two experiments clearly support- that the vesicular LAT and VAMP7 vesicle function is clearly associated with attenuation of micro-cluster signaling.

Major points

1. Experiments in Figure 1 show that buffering of Calcium and inhibition of microtubules both results in increase microcluster phosphorylation and size. The authors state fail to state that this results is perfectly inconsistent with their initial or revised working model and just state that they can't explain it. They come back to the issue of signal termination at the end, but otherwise don't connect this explicitly to their own results. In the JI paper the authors examined truncated LAT constructs, but they have not looked at the mutants reported by Kunii et al to determine if reduction of ubiquitination would change LAT distribution and have similar effects on microcluster dynamics.
2. The tomographic reconstructions in Figure 4 simplifies the scoring of vesicular structures to dots, but should have actually shown images of reconstructed cytoplasmic membrane system. While the need to cover large volume has somewhat degraded their resolution to 10 nm, running the movies makes it clear that in addition to small vesicles there are also tubulovesicular structures with intraluminal vesicles that may even be mis-scored as being part of smaller vesicles. These structures are expected near the interface, during TCR signaling, but its not clear how much LAT is contributing to these are the CLEM cannot separate signals from MVB from signals from other vesicles that are short distances away. Nonetheless, complete reconstructions that trace the membranes and model the connection in 3D should be shown and MVB and small transport vesicles both quantified.
3. Many aspects of the analysis of the LAT and VAMP7 positive structures in relation to TCR is very similar to the analysis carried out by Purbhoo et al. The new imaging technology provides greater time-resolution and ability to confirm that relationship to microtubules, which was already supported by some data. The main new observation is the ability to see the LAT flashes where the VAMP7 vesicles contact with microclusters. However, its otherwise not followed beyond description. Again, it seems that there is a new descriptive finding that is not followed up or exploited to obtain a deeper understanding beyond the prevailing model.

Reviewer #3 (Remarks to the Author):

In their paper entitled "Plasma membrane LAT recruitment precedes vesicular LAT recruitment to reveal two phases of early T cell activation", Lakshmi Balagopalan and co-workers address the still debated question of the mechanisms of recruitment of the adaptor protein LAT. Using high resolution microscopy, the authors claim that there are two phases in the formation of LAT clusters at the immune synapse. The first phase (less than 5 min after activation) does not require vesicular transport. The second phase is characterized by a polarized movement of vesicles containing LAT and VAMP7, which move between the already formed LAT microclusters. This study is contributing to a debate on the mechanisms of formation of LAT clusters during activation of T lymphocytes. It is novel in the sense that the authors by using lattice sheet microscopy and TIRFM-SIM characterize precisely the two phases of recruitment of LAT at the immune synapse. It is certainly important for scientists in the field since it reconciles two interpretations of the data already published: The supporters of a mechanisms of formation of LAT clusters relying only on the pool of LAT already present at the plasma membrane and those of a formation only dependent on a recruitment of a vesicular pool. The data are convincing and well performed.

Arguments for the publication:

As stated before this study answers a still open debate on the formation of LAT clusters at the immune synapse.

The experiments are well performed and convincing. They are novel in the sense they show for the first time the precise kinetic formation of the clusters containing LAT.

Major point to be answered:

Although the study is very elegant and well performed it mainly characterizes the two phases of recruitment of LAT but does not correlate these two phases with the two phases of early T cell activation the authors mention in the title of their study. They should try to document the two phases of LAT recruitment described in their study in terms of activation. They could do that by analyzing more thoroughly at different time points the recruitment of signaling molecules to the activation sites. Indeed, only Grb2 is shown in Figure 3, and this recruitment is not analyzed kinetically. Do all signaling molecules known to associate with LAT form cluster in the first phase of recruitment? What happen to phospho-LAT microclusters and other signaling clusters during the second phase? What happen to the formation of these microclusters in cells treated with nocodazole at early AND LATER time points? The authors could also measure phosphor-Erk in the different conditions.

A kinetic analysis of the recruitment of the activating molecules and/or of their phosphorylation would contribute to answer the still unresolved question, which is: What is the contribution of each pool of LAT in T cell activation?

Specific comments:

Figure 1:

The authors use a chelator of Ca²⁺ and show that the phosphorylation of LAT and CD3z at the immune synapse are not decreased but increased. Their interpretation is that although the fusion of vesicles at the immune synapse is blocked the activation is not altered. Yet in their conditions, the authors do not show that there is no fusion. Moreover, to argue that LAT containing vesicles recruited to the immune synapse fuse with the plasma membrane they wrongly cite the article from Larghi et al. (reference 14), which precisely shows that there is no fusion. The authors measure the phosphorylation of LAT 2 minutes after synapse formation, whereas as shown later, this time point is a little too early for the recruitment of all the intracellular vesicles. This analysis should thus also be performed at later time points. Also, the authors should show both cluster area and cluster density in the different conditions used. Indeed, use of drugs may also alter the spreading of Jurkat cells on anti-CD3 coated slides.

Figure 2:

The authors also use nocodazole treatment to block the polymerization of microtubules and the recruitment of vesicles to the synapse. They show that this treatment does not diminish the formation of pLAT microclusters at the plasma membrane. Again, this experiment is done 2 minutes after dropping. It would be important to see if at later time points there is a difference in LAT and pLAT clusters at the synapse and if it alters the formation of other microclusters (Grb2, SLP76, PLCg1...). Time of first appearance of these clusters should be shown.

The authors show very nicely that VAMP7 vesicles contact LAT clusters already present at the immune synapse and that this induces "flares" of LAT that can be either due to fusion of VAMP7 vesicle or co-clustering of the vesicular LAT in trans with plasma membrane LAT pool. Statistical analysis of the time of residency of VAMP7 at the synapse and of the loss of mobility of LAT cited line 8 of page 18 would contribute to better characterize these events.

Minor comments:

The authors should show the separated channels for all the movies in supplemental data.

Reviewers' comments:

Reviewer #1 (Remarks to the Author):

In addition to its localization at the plasma membrane of resting T cells, Lat also resides on intracellular vesicles. Analysis of T cells deficient in VAMP-7, suggested that the docking of Lat-containing intracellular vesicles is mandatory for proper TCR signaling. However, given that Lat molecules localized to the plasma membrane (PM) are efficiently phosphorylated in the seconds that follow engagement of the TCR, it has been argued that vesicular Lat molecules are required to sustain signaling only at late time points. Therefore, although indisputable experimental evidence supports the proposal of the existence of a pool of vesicular Lat molecules, contrasting views remain as to its exact function. The present study aims at solving this controversy by assessing the dynamics of the two existing pools of Lat molecules. Using pharmacological agents (BAPTA/EGFTA and nocodazole), the authors first established that neither vesicle recruitment nor calcium-mediated exocytosis are required for LAT microcluster formation at the PM, suggesting that the latter occurs independently of VAMP7-associated vesicles. To define the relationship between the pool of Lat found at PM and associated with intracellular vesicles and their relative contribution to Lat microcluster formation, they used lattice light sheet microscopy (LLSM) to simultaneously image Lat and a vesicular marker VAMP7-positive vesicles. They demonstrated that there exists a lag of 45-60 s between LAT microcluster formation at the PM and recruitment of the VAMP7-positive vesicles to the synapse. A finding further supported by Total Internal Reflection Fluorescence Simulated Interference Microscopy using Lat, Grb2, Zap70 and VAMP-7 reporters and by correlative 3D light and Focused Ion Beam Scanning Electron Microscopy. Although this study does not provide a final answer concerning the role played by the Lat-containing intracellular vesicles that dock at the immunological synapse 45-60 seconds after initiation of activation, the present study that capitalizes on the most advanced imaging tools provides a novel and very precise view on the dynamics of the two pools of Lat. Moreover, the dynamics relationships that exist between the VAMP7-positive vesicles and the PM areas that contained ZAP-70 microclusters, the coincidence of VAMP7 vesicle movement and microtubules, and the enhanced LAT fluorescence that occurs when a VAMP7-positive vesicle approached and made contact with the activated T cell surface are particularly fascinating. The discussion fairly analyzes these novel and exciting results.

We thank the reviewer for accurately summarizing our results and the positive comments.

Specific (minor) comments

1/ In Fig. 4Ai, ROI 2, the spotted VAMP7-positive vesicle located further above in the cell volume appears associated with pLAT. This is bit puzzling since no pLAT should be present so far away from the stimulatory coverslip. The authors should comment on that issue.

We believe that due to the 4 colors in the image, the pLAT signal may have appeared to this Reviewer to be in ROI2 of Fig. 4Ai. We have now included panels showing the isolated red (pLAT) and green (VAMP7-emerald) channels of ROI 1 and 2 of both Fig. 4A and B in Fig. S3C. It is clear from these panels that ROI2 of Fig. 4Ai contains only green VAMP7-emerald signal and no pLAT signal.

2/ It will be interesting to know the extent of T cell activation that occurs during the first 30-60 s of activation prior to relocalization of the pool of vesicular LAT at the IS. For instance, using the Jurkat model at what time point is it possible to monitor the onset of ERK phosphorylation? Along the same line is ERK phosphorylated after nocodazole treatment?

We thank the reviewer for this insightful comment and in the revised version of the manuscript we have characterized T cell activation before and after vesicle recruitment. We have done this in 2 ways: kinetic analysis of microcluster composition by microscopy (Fig. 7) and phosphorylation kinetics of signaling proteins by biochemistry (Fig. S1 C and D).

In Fig. 7 the kinetics of T cell activation induced recruitment of LAT and LAT-associated signaling proteins to microclusters before and after vesicle recruitment is described. LAT and LAT-associated proteins Grb2, PLC γ 1 and SLP76 are present in microclusters before vesicular recruitment. Coincident with VAMP7 vesicle recruitment, the fluorescence of the signaling molecules increased, consistent with a role for VAMP7 vesicles in microcluster maintenance and sustained signaling.

We have also analyzed early and late phosphorylation kinetics of these signaling molecules by western blot (Fig. S1 C and D). It appears that these signaling molecules are phosphorylated as early as 10 seconds after CD3 stimulation (the earliest time-point evaluated), well before vesicle recruitment, and phosphorylation peaks at 2 mins as previously reported by Houtman et al. ¹. As the reviewer requested, we analyzed ERK phosphorylation upon TCR stimulation. We could detect phospho-ERK as early as 10 sec after CD3 stimulation. Surprisingly, though nocodazole treatment resulted in decreased signaling in LAT and LAT-associated proteins at later time points, ERK phosphorylation was elevated in nocodazole-treated cells (Fig. S1D). Our results are consistent with previous reports in which microtubule-interfering agents stimulate MAPK superfamily members and one study in particular in which a nocodazole-induced increase in phospho-ERK was reported ².

3/ If I understand properly the reasoning of the authors, they suggest that in the experiments reported by Hivroz and colleagues, deletion of VAMP7 is 'chronic' and might adventitiously affect the constitution of the pool of Lat found at the PM. Since VAMP7 is of much interest for neurobiologists, is there any drugs or optogenetics devices permitting an 'acute' ablation of VAMP7?

We thank the reviewer for this suggestion and we have looked extensively to the neurobiology literature to address this very issue. Usually, toxins (e.g. botulinum and tetanus toxin) that are potent and selective proteases, which cleave v-SNAREs are used to inhibit synaptic vesicle exocytosis and neurotransmitter release ³. Tetanus toxin was also used in T cells to inhibit TCR accumulation at the immune synapse ⁴. However, VAMP7 is also known as Ti-VAMP or tetanus toxin insensitive VAMP. So we have had to look at other approaches to inhibit VAMP7 function. We used BAPTA and nocodazole in Figure 1 as general inhibitors of exocytosis and vesicle recruitment respectively and saw no difference in LAT microcluster formation.

In response to this reviewer's comment for VAMP-7 specific acute inhibition, we generated a genetically engineered rapamycin-induced dimerization system. In this system two proteins are genetically fused to FKBP12 and FRB and are brought into close proximity in the presence of

rapamycin. We generated a system to sequester VAMP7 vesicles away from the synapse and to the mitochondria. This technique, called “knock-sideways”, has been used to inactivate clathrin adapter proteins and ADP-ribosylation proteins by forced translocation to mitochondria^{5,6}. Clathrin adapter protein AP-2 was sequestered to the mitochondria as rapidly as 3 seconds upon application of rapamycin⁶.

We constructed FKBP-VAMP7 and expressed it in combination with AKAP-FRB, a protein that localizes to the mitochondrial outer membrane (kindly provided to us by Dr. Tamas Balla). We hypothesized that rapamycin-induced heterodimerization would induce translocation of VAMP7 vesicles to the mitochondria. In the absence of rapamycin, there was no colocalization between the two constructs (Response to Reviewers Figure 1A ii). Analysis of the time course of addition of rapamycin revealed that it took 15 min for VAMP7 to be almost completely rerouted to mitochondria (Response to Reviewers Figure 1B ii). We were surprised by the long incubation time required for translocation of VAMP7 vesicles, but this is probably because the vesicles need to physically contact the mitochondrial membrane to be sequestered. After 15 minutes of rapamycin treatment, early microcluster formation was intact, but sustained microclusters disappeared faster compared with untreated cells (Response to Reviewers Fig. 1 compare A and B). However, control cells treated with rapamycin, but not transfected with AKAP-FRB, also showed faster microcluster disappearance (Response to Reviewers Fig. 1C). Thus in our system, the effect of rapamycin alone for the length of time required for translocation of VAMP7 vesicles to the mitochondria, precludes the interpretation of these results. Our future plans include exploring optogenetic methods to acutely inhibit VAMP7 function and also selectively interrogate the function of cell surface versus intracellular LAT.

Reviewer #2 (Remarks to the Author):

The group recently published a study in JI that makes a strong case that plasma membrane LAT clusters transiently with TCR, is phosphorylated at TCR microclusters and makes up 90% of phosphorylated LAT measured at 2 minutes. This seemed to leave a small potential role for phosphorylation of LAT that is intracellular at the onset of activation, making up approximately 40% of cellular LAT. This study uses state of the art 3D and eTIRF-SIM imaging approaches to further evaluate the role of sub-synaptic vesicles containing LAT in initiation of TCR signaling. They also include an analysis of VAMP-7 containing vesicles, which carry a fraction of the intracellular LAT and could deliver it to the plasma membrane in a Ca²⁺ dependent manner. The study supports the conclusion of the earlier work in JI that plasma membrane LAT is the main pool use early signaling. The first two experiments in the paper, one involving Calcium flux buffering and another involving microtubule

depolymerization, both immediately point to a problem with control of TCR microcluster growth or signal termination. The CLEM analysis further demonstrates that vesicular compartments near the synapse include many tubulovesicular structures with intraluminal vesicles, which are known as multivesicular bodies and are involved in endosome to lysosome trafficking of ubiquitinated membrane proteins. In fact, LAT is known to be ubiquitinated and when these lysine are changed to arginines TCR signaling is positively disregulated (Kunii N, Zhao Y, Jiang S, Liu X, Scholler J, Balagopalan L, Samelson LE, Milone MC, June CH. Enhanced function of redirected human T cells expressing linker for activation of T-cells that is resistant to ubiquitylation. Hum Gene Ther. 2012. Epub 2012/09/25. doi: 10.1089/hum.2012.130. PubMed PMID: 22998346.). The authors may

already recognize these signatures as the discussion clearly cites some relevant papers regarding endocytosis, signal termination through ESCRT function, etc, but the descriptive components of the paper fail to call attention to these features. The restate an earlier hypothesis first proposed by Purbhoo et al and Soares et al, that the vesicular LAT delivery is important for sustained signaling. My feeling is that the current paper generates no new conclusions, but at the same time is full of missed opportunities to recognize its own novel results and design new experiments that test the novel initial results. At present it would have a low priority based on the lack of any conceptual novelty is written, but a fantastic potential if it were to put together the obvious new thread the authors have uncovered some targeted experiments to test the new hypothesis that the first two experiments clearly support- that the vesicular LAT and VAMP7 vesicle function is clearly associated with attenuation of micro-cluster signaling.

We appreciate the Reviewer's description of our previous results ⁷. We note however that the field has not fully accepted our results. The discussion of Soares et al., J Exp. Med ⁸ questions the use of CD4-LAT as a tool and several reviews still discuss the controversy of the relative roles of surface versus intracellular LAT in initiating T cell signaling. This is why we felt it imperative that we address and resolve the controversy using new imaging technologies available to us.

We also thank the Reviewer for the comments. The Reviewer brings up an important point about a possible role for vesicles in signal termination as indicated by BAPTA and nocodazole inhibition in our original version of Figure 1. Please look at our response to Major Point 1 below.

Major points

1. Experiments in Figure 1 show that buffering of Calcium and inhibition of microtubules both results in increase microcluster phosphorylation and size. The authors state fail to state that this results is perfectly inconsistent with their initial or revised working model and just state that they can't explain it. They come back to the issue of signal termination at the end, but otherwise don't connect this explicitly to their own results.

The Reviewer brings up a valid point. However, in the time when this manuscript was under review, we discovered that most of the phospho-LAT antibodies that we had been using were unfit for immunofluorescence (see Response to Reviewers Fig. 2 below). We tested them in a recently generated LAT KO CRISPR Jurkat cell line (kindly provided by Dr. Art Weiss) and concluded that pLAT226 was the only antibody that we could use for quantitative immunofluorescence. So we repeated the BAPTA and nocodazole experiments in Figure 1 using pLAT226. As described in the Results section describing Figure 1, we do not see a significant increase in pLAT cluster area or intensity at early time points in BAPTA and nocodazole treated cells. We have also included analysis of later time-points and BAPTA and nocodazole show divergent results. pLAT microcluster size and intensity were slightly increased at later times in BAPTA treated cells, but significantly decreased in nocodazole treated cells. This result is consistent with a previously reported role for MTOC repositioning for sustained T cell signaling ⁹. We also observed consistent results in western blots using phospho antibodies to LAT, PLC γ 1 and SLP-76 (Fig. S1 C and D). Thus our results suggest that neither vesicle exocytosis nor recruitment are required for LAT microcluster formation. The increase in pLAT clusters observed in BAPTA-treated cells at later times is the topic of another study in our lab that investigates the role of calcium in proximal TCR signaling.

Moreover, the revised version of the manuscript now includes a kinetic analysis of T cell activation induced recruitment of LAT and LAT-associated signaling proteins to microclusters before and after vesicle recruitment in Figure 7. LAT and LAT-associated proteins Grb2, PLC γ 1 and SLP76 are present in microclusters before vesicular recruitment. Coincident with VAMP7 vesicle recruitment, the fluorescence of LAT and LAT-binding signaling molecules increase, consistent with a role for VAMP7 vesicles in microcluster maintenance.

Taken together, the decrease in sustained signaling in nocodazole treated cells as well as the increase in microcluster intensity coincident with VAMP7 vesicle recruitment are consistent with a role for VAMP7 vesicles in signal sustenance and not signal termination.

In the JI paper the authors examined truncated LAT constructs, but they have not looked at the mutants reported by Kunii et al to determine if reduction of ubiquitination would change LAT distribution and have similar effects on microcluster dynamics.

We refer this reviewer to our paper in PNAS in which we reported on the localization and microcluster dynamics of the ubiquitination-defective LAT 2KR mutant¹⁰. This is the same mutant reported by Kunii et al. Briefly, 2KRLAT microcluster dynamics were unchanged compared with WTLAT. Studies are currently underway to look at cellular distribution and cellular trafficking of this mutant. Preliminary results indicate that endocytosis of 2KR LAT compared with WT LAT is unaltered, but cellular trafficking is changed.

2. The tomographic reconstructions in Figure 4 simplifies the scoring of vesicular structures to dots, but should have actually shown images of reconstructed cytoplasmic membrane system. While the need to cover large volume has somewhat degraded their resolution to 10 nm, running the movies makes it clear that in addition to small vesicles there are also tubulovesicular structures with intraluminal vesicles that may even be mis-scored as being part of smaller vesicles. These structures are expected near the interface, during TCR signaling, but its not clear how much LAT is contributing to these are the CLEM cannot separate signals from MVB from signals from other vesicles that are short distances away. Nonetheless, complete reconstructions that trace the membranes and model the connection in 3D should be shown and MVB and small transport vesicles both quantified.

In response to this reviewer's comments, the FIB-SEM subvolumes corresponding to Fig. 4 were re-segmented manually to highlight the tubulovesicular structures in the cell and are now included in Fig. S3 A and B. In addition to the nucleus and cytosol already shown in Fig. 4 (red and yellow respectively), membranes in these volumes were segmented based on their contrast in the FIB-SEM images and were false colored green. In addition, there was a mitochondrion captured in 4Aviii; this was also segmented and colored blue. Because these features obscured the small cluster of vesicles, the vesicles were highlighted as purple spheres.

3. Many aspects of the analysis of the LAT and VAMP7 positive structures in relation to TCR is very similar to the analysis carried out by Purbhoo et al. The new imaging technology provides greater time-resolution and ability to confirm that relationship to microtubules, which was already supported by some data. The main new observation is the ability to see the LAT flashes where the VAMP7 vesicles contact with microclusters. However, its otherwise not followed beyond

description. Again, it seems that there is a new descriptive finding that is not followed up or exploited to obtain a deeper understanding beyond the prevailing model.

In order to obtain a greater understanding of the novel LAT “flares” we observed on the lattice light sheet microscope, we applied for and obtained time on the LLSM at Janelia Farms. We performed experiments there for three weeks from 9/16-10/7. However, due to several microscope-related technical difficulties on their LLSM, we were unsuccessful in getting meaningful data from the time we spent there.

However, in response to the Reviewer’s comment, we have included in the revised version of the manuscript, new analysis of the “flares” from the data we already had. We have now included in Fig. 6Bv, analysis of the speed of the VAMP7 vesicle away from the “flare event” and at the flare event. We see a significant difference in the average speed of VAMP7 vesicles away from the flare ($0.3478 \pm 0.0285 \mu\text{m}/\text{sec}$) and during a flare ($0.1137 \pm 0.008458 \mu\text{m}/\text{sec}$), consistent with a loss of mobility of the vesicle and what appears to be “trapping” of the vesicle at the flare.

Reviewer #3 (Remarks to the Author):

In their paper entitled “Plasma membrane LAT recruitment precedes vesicular LAT recruitment to reveal two phases of early T cell activation”, Lakshmi Balagopalan and co-workers address the still debated question of the mechanisms of recruitment of the adaptor protein LAT. Using high resolution microscopy, the authors claim that there are two phases in the formation of LAT clusters at the immune synapse. The first phase (less than 5 min after activation) does not require vesicular transport. The second phase is characterized by a polarized movement of vesicles containing LAT and VAMP7, which move between the already formed LAT microclusters.

This study is contributing to a debate on the mechanisms of formation of LAT clusters during activation of T lymphocytes. It is novel in the sense that the authors by using lattice sheet microscopy and TIRFM-SIM characterize precisely the two phases of recruitment of LAT at the immune synapse. It is certainly important for scientists in the field since it reconciles two interpretations of the data already published: The supporters of a mechanisms of formation of LAT clusters relying only on the pool of LAT already present at the plasma membrane and those of a formation only dependent on a recruitment of a vesicular pool. The data are convincing and well performed.

Arguments for the publication:

As stated before this study answers a still open debate on the formation of LAT clusters at the immune synapse.

The experiments are well performed and convincing. They are novel in the sense they show for the first time the precise kinetic formation of the clusters containing LAT.

We thank the reviewer for accurately summarizing our results and the positive comments.

Major point to be answered:

Although the study is very elegant and well performed it mainly characterizes the two phases of recruitment of LAT but does not correlate these two phases with the two phases of early T cell activation the authors mention in the title of their study. They should try to document the two

phases of LAT recruitment described in their study in terms of activation. They could do that by analyzing more thoroughly at different time points the recruitment of signaling molecules to the activation sites. Indeed, only Grb2 is shown in Figure 3, and this recruitment is not analyzed kinetically. Do all signaling molecules known to associate with LAT form cluster in the first phase of recruitment? What happen to phospho-LAT microclusters and other signaling clusters during the second phase? What happen to the formation of these microclusters in cells treated with nocodazole at early AND LATER time points? The authors could also measure phosphor-Erk in the different conditions.

A kinetic analysis of the recruitment of the activating molecules and/or of their phosphorylation would contribute to answer the still unresolved question, which is: What is the contribution of each pool of LAT in T cell activation?

We thank the reviewer for this very insightful comment and in the revised version of the manuscript we have characterized T cell activation before and after vesicle recruitment. We have done this in 2 ways: by analyzing microcluster composition microscopically (Fig. 7) and phosphorylation kinetics by biochemistry (Fig. S1 C and D).

In Fig. 7 the kinetics of T cell activation induced recruitment of LAT and LAT-associated signaling proteins to microclusters before and after vesicle recruitment is described. LAT and LAT-associated proteins Grb2, PLC γ 1 and SLP76 are present in microclusters before vesicular recruitment. Coincident with VAMP7 vesicle recruitment, the fluorescence of the signaling molecules increased, consistent with a role for VAMP7 vesicles in microcluster maintenance and sustained signaling.

We have also analyzed early and late phosphorylation kinetics of these signaling molecules by western blot (Fig. S1 C and D). It appears that these signaling molecules get phosphorylated as early as 10 seconds after CD3 stimulation, well before vesicle recruitment, and phosphorylation peaks at 2 mins as previously reported by Houtman et al. ¹. As the reviewer requested, we detected ERK phosphorylation upon TCR stimulation. We could detect phosphor-ERK as early as 10 sec after CD3 stimulation prior to the timeframe in which vesicles are recruited. Surprisingly, though nocodazole treatment resulted in decreased signaling in LAT and LAT-associated proteins at later time points, ERK phosphorylation was elevated in nocodazole treated cells (Fig. S1D). Our results are consistent with previous reports in which microtubule-interfering agents stimulate MAPK superfamily members and one study in particular in which nocodazole-induced increase in phospho-ERK was reported ².

Specific comments:

Figure 1:

The authors use a chelator of Ca²⁺ and show that the phosphorylation of LAT and CD3z at the immune synapse are not decreased but increased. Their interpretation is that although the fusion of vesicles at the immune synapse is blocked the activation is not altered. Yet in their conditions, the authors do not show that there is no fusion.

For regulated exocytosis, it is well established in neuronal synapses that elevation of Ca²⁺ is required, which could be due to either Ca²⁺ entry across the plasma membrane or Ca²⁺ mobilization from internal stores or both. Moreover, in T cells, Soares et al. JEM 2013 used BAPTA to prevent fusion of synaptotagmin VII vesicles and reported that calcium chelation prevented LAT and TCR ζ

microcluster formation (reported as unpublished data⁸). In our experiments we used BAPTA and EGTA to chelate calcium both outside and inside the cell. Thus we reasoned that calcium chelation using BAPTA and EGTA should prevent exocytosis. In revised Figure 1 we show that phospho-LAT cluster formation is unaffected by calcium chelation, data that contradict the results reported in Soares et al.

Moreover, to argue that LAT containing vesicles recruited to the immune synapse fuse with the plasma membrane they wrongly cite the article from Larghi et al. (reference 14), which precisely shows that there is no fusion.

On rereading this sentence, we can see what the reviewer means and we have removed this reference from this particular sentence.

The authors measure the phosphorylation of LAT 2 minutes after synapse formation, whereas as shown later, this time point is a little too early for the recruitment of all the intracellular vesicles. This analysis should thus also be performed at later time points. Also, the authors should show both cluster area and cluster density in the different conditions used. Indeed, use of drugs may also alter the spreading of Jurkat cells on anti-CD3 coated slides.

The reviewer makes a valid point and we have now measured phospho-LAT cluster area and intensity at both early and later time-points in DMSO and BAPTA treated cells. These results are now included in Figure 1. We have also analyzed early and late phosphorylation kinetics of LAT and LAT-associated signaling molecules Grb2, PLC γ 1 and SLP76 in DMSO and BAPTA-treated cells by western blot (Fig. S1 C and D)

We would like to inform this reviewer that in the time when this manuscript was under review, we discovered that most of the phospho-LAT antibodies that we had been using were unfit for immunofluorescence (see Response to Reviewers Fig. 2). We tested them in a recently available LAT KO CRISPR Jurkat cell line and concluded that pLAT226 was the only one that we could use for quantitative immunofluorescence. So we repeated the BAPTA and nocodazole experiments in Figure 1 using pLAT226.

As written in the Results section describing Figure 1, we do not see a significant increase in pLAT cluster area or intensity at early time points in BAPTA treated cells, but a slight increase in pLAT clusters at later times. Thus our results suggest that neither vesicle exocytosis nor recruitment are required for LAT microcluster formation. The increase in pLAT clusters observed in BAPTA-treated cells at later times is the topic of another study in our lab that investigates the role of calcium in proximal TCR signaling. We observed consistent results in western blots using phospho antibodies to LAT, PLC γ 1 and SLP-76, results of which are included in Fig S1 C and D.

Figure 2:

The authors also use nocodazole treatment to block the polymerization of microtubules and the recruitment of vesicles to the synapse. They show that this treatment does not diminish the formation of pLAT microclusters at the plasma membrane. Again, this experiment is done 2 minutes after dropping. It would be important to see if at later time points there is a difference in LAT and

pLAT clusters at the synapse and if it alters the formation of other microclusters (Grb2, SLP76, PLCγ1...). Time of first appearance of these clusters should be shown.

The reviewer makes a valid point and we have now measured phospho-LAT cluster area and intensity at both early and later time-points in DMSO and nocodazole treated cells. These results are now included in Figure 1. As written in the Results section describing Figure 1, we do not see a significant change in pLAT cluster area or intensity at early time points in nocodazole-treated cells. However at later times we observed significantly decreased phospho-LAT cluster area and intensity, consistent with a role for vesicle recruitment in sustained signaling. This result is consistent with a previously reported role for MTOC repositioning for sustained T cell signaling⁹. We also observed consistent results in western blots using phospho antibodies to LAT, PLCγ1 and SLP-76, results of which are included in Fig S1 C and D.

The authors show very nicely that VAMP7 vesicles contact LAT clusters already present at the immune synapse and that this induces “flares” of LAT that can be either due to fusion of VAMP7 vesicle or co-clustering of the vesicular LAT in trans with plasma membrane LAT pool. Statistical analysis of the time of residency of VAMP7 at the synapse and of the loss of mobility of LAT cited line 8 of page 18 would contribute to better characterize these events.

We thank the reviewer for this suggestion and have now included in Fig. 6Bv analysis of the speed of the VAMP7 vesicle away from the “flare event” and at the flare event. We see a very significant difference in the average speed of VAMP7 vesicles away from the flare ($0.3478 \pm 0.0285 \mu\text{m}/\text{sec}$) and during a flare ($0.1137 \pm 0.008458 \mu\text{m}/\text{sec}$), consistent with a loss of mobility of the vesicle and what appears to be “trapping” of the vesicle at the flare.

REFERENCES

1. Houtman, J.C., Houghtling, R.A., Barda-Saad, M., Toda, Y. & Samelson, L.E. Early phosphorylation kinetics of proteins involved in proximal TCR-mediated signaling pathways. *J Immunol* **175**, 2449-2458 (2005).
2. Guo, X. *et al.* Nocodazole increases the ERK activity to enhance MKP-1 expression which inhibits p38 activation induced by TNF-alpha. *Mol Cell Biochem* **364**, 373-380 (2012).
3. Schiavo, G., Matteoli, M. & Montecucco, C. Neurotoxins affecting neuroexocytosis. *Physiol Rev* **80**, 717-766 (2000).
4. Das, V. *et al.* Activation-induced polarized recycling targets T cell antigen receptors to the immunological synapse; involvement of SNARE complexes. *Immunity* **20**, 577-588 (2004).
5. Hirst, J. *et al.* Distinct and overlapping roles for AP-1 and GGAs revealed by the “knocksideways” system. *Curr Biol* **22**, 1711-1716 (2012).

6. Robinson, M.S., Sahlender, D.A. & Foster, S.D. Rapid inactivation of proteins by rapamycin-induced rerouting to mitochondria. *Dev Cell* **18**, 324-331 (2010).
7. Balagopalan, L., Barr, V.A., Kortum, R.L., Park, A.K. & Samelson, L.E. Cutting edge: cell surface linker for activation of T cells is recruited to microclusters and is active in signaling. *J Immunol* **190**, 3849-3853 (2013).
8. Soares, H. *et al.* Regulated vesicle fusion generates signaling nanoterritories that control T cell activation at the immunological synapse. *J Exp Med* **210**, 2415-2433 (2013).
9. Martin-Cofreces, N.B. *et al.* MTOC translocation modulates IS formation and controls sustained T cell signaling. *J Cell Biol* **182**, 951-962 (2008).
10. Balagopalan, L. *et al.* Enhanced T-cell signaling in cells bearing linker for activation of T-cell (LAT) molecules resistant to ubiquitylation. *Proc Natl Acad Sci U S A* **108**, 2885-2890 (2011).

Response to Reviewers Figure 1

Response to Reviewers Figure 1: FKBP-FRB system to sequester VAMP7 vesicles to mitochondria.

A and B. Jurkat cells were transfected with RFP-FKBP-VAMP7, Grb2-halo and AKAP-FRB-CFP.

Following Halo labeling of Grb2, cells were either treated with DMSO (-Rapa) or 100nm Rapamycin (+Rapa) for 15 minutes and then dropped onto stimulatory coverslips and imaged by TIRF microscopy.

A. i. Time course of Grb2 cluster formation, followed by VAMP7 vesicle recruitment in DMSO treated cells. **ii.** Upper panel shows epifluorescence image of AKAP-FRB-CFP showing mitochondrial localization, while the lower panel shows epifluorescence image of FKBP-VAMP7 localized to 2 pools.

B. i. Time course of Grb2 cluster formation, followed by VAMP7 vesicle recruitment in rapamycin treated cells. **ii.** Upper panel shows epifluorescence image of AKAP-FRB-CFP showing mitochondrial localization, while the lower panel shows epifluorescence image of FKBP-VAMP7 sequestered to mitochondria.

C. Jurkat cells were transfected with RFP-FKBP-VAMP7 and Grb2-halo in the absence of AKAP-FRB-CFP. Following Halo labeling of Grb2, cells were either treated with 100nm Rapamycin (+Rapa) for 15 minutes and then dropped onto stimulatory coverslips and imaged by TIRF microscopy. **i.** Time course of Grb2 cluster formation and VAMP7 vesicle recruitment in rapamycin treated cells.

Response to Reviewers Figure 2

Response to Reviewers Figure 2: phospho-LAT 226 is the only phospho-LAT antibody fit for quantitative immunofluorescence. **A.** WT Jurkat and JLATKO cells were dropped onto stimulatory coverslips and fixed after 3 minutes of stimulation. The cells were then immunostained with the indicated pLAT antibodies and imaged by TIRF microscopy. **B.** Graph of fluorescence intensity detected by pLAT antibodies in WT and JLATKO cells.

Reviewers' comments:

Reviewer #1 (Remarks to the Author):

None

Reviewer #2 (Remarks to the Author):

The authors have addressed my concerns and those of other reviewers. The authors have made significant refinements and the general appearance that vesicles appearance corresponded to signal termination in micro clusters are less striking after the revisions and the role of Calcium will be pursued in future studies. I think this is reasonable given the amount of data in the current manuscript. They have also attempted to extend the LLS studies using additional data they had in hand, which improve the depth of analysis. They have also undertaken a significant manual segmentation of EM data to better demarcate different types of membrane structures. The authors also performed a more detailed analysis of the LAT flashes. I accept the authors arguments that the issues are still contentious and that the analysis here will further clarify the trafficking of lat in time frames of second and minutes following triggering of TCR signaling.

Reviewer #3 (Remarks to the Author):

General comment:

One of the main comment from the three reviewers was that it would be important to analyze the extent of T cell activation that occurs during the first 30-60 s of activation prior to relocalization of the pool of vesicular LAT at the IS in order to relate that to the 2 phases of T cell activation the authors are eluding to in their title.

Although the authors have made new experiments to answer this point, the results of the experiments shown in the paper do not allow to draw the conclusions made by the authors.

Specific comments:

Indeed, the microscopy analysis of the recruitment of signaling molecules presented in Figure 7, only show one cell for each signaling molecule studied. Several images should be analyzed and quantified rigorously to draw any conclusion. It is particularly important since it seems that at 0 Grb2 and SLP76 are already associated with Lat, which raises the following question "is t0 really a t0?" and thus the all problem of the kinetic.

It is also very striking that VAMP7 in blue in the figure is seen in the early time point (48 sec), which somehow contradicts what the authors wrote about the kinetic of recruitment of the VAMP7-positive vesicular pool of Lat. Finally the figure as it is, is difficult to interpret since time scale is not clearly indicated in 7D, E and F (Does t14 correspond to 14sec? What is the scale in total fluorescence intensity representation in right panels?). For the WB analysis of the signaling in activated Jurkat T cells, the authors should also performed a quantification of several experiments to present a statistical analysis of the kinetic of activation. It is particularly important since activation pattern and kinetic seems variable in the 2 experiments presented in the figure: compare DMSO conditions in S1C and S1D for phospho-SLP76 and phospho-PLCg1. It is also striking that little phosphorylation of LAT is observed before 30 seconds in figure S1C (quantification shows a 3.3 increase at 10sec and 2.8 at 30 seconds for phospho-Lat but this seems overestimated when compared to blot below showing 2.9 and 2.4 increase for phospho-SLP76 at the same time) and D (No increase of phospho-LAT at 10 and 15 seconds, 4.5 fold at 2 min). This is in contradiction with the fact that the plasma membrane pool of LAT would mostly contribute to activation. The peak at 2 minutes may indeed correspond to the recruitment of the vesicular pool suggesting that signaling requires recruitment of vesicular pool.

Concerning the results obtained with BAPTA and nocodazole they are difficult to interpret:

- For Western blots, because of a lack of quantification of several experiments and of the

variability of the 2 experiments presented. No effect of nocodazole early on because no early activation of phospho-Lat and phospho-PLC γ in DMSO but clear effect on phospho-SLP76 as early as 10 sec in the one experiment presented. It also seems that there is less phospho-LAT at 2 minutes in figure 1E. Thus, the conclusion of the authors that nocodazole does not affect early signaling in T cells is in contradiction with the results they show.

- For microscopy images, quantification of images only for phospho-Lat and not for phospho-Tyr (which would give a more global idea of signaling in T cells). Moreover, it seems that the distribution of dots in 1B and perhaps 1C is not normal (bimodal). Is the statistical analysis used appropriate for such a distribution? Although authors show phospho-Lat and total Lat in 1D they do not quantify total LAT, which would also be important to document.

- The interpretation of the results on BAPTA by the authors is not stated clearly enough. The results indeed clearly rule out the fact that exocytosis of the vesicles containing Lat at the immune synapse is required for phosphorylation and recruitment of LAT at 5 minutes. Thus the docking model of LAT vesicles just below the plasma membrane without fusion is the correct model.

MEDIATING COMMENTS ON REF3 REPORT

Reviewer 1 says:

I went through the comments of Reviewer 3 and through the revised MS. It sounds to me that the most important issue to be fixed corresponds to the t_0 issue in Figure 7 ('it is particularly important since it seems that at t_0 Grb2 and SLP76 are already associated with Lat, which raises the following question "is t_0 really a t_0 ?" and thus the all problem of the kinetic'). All the other issues can be readily fixed by providing the requested refined data. The authors were also very cautious in their interpretation of the data. I do however agree with Reviewer 3 in that the docking model of LAT vesicles just below the plasma membrane without fusion is likely the correct model. I still consider the present paper as a major contribution to the field and suggest to have it published in the very near future.

Reviewer 2 says:

I think the paper should be accepted, but the authors should respond as best they can to the comments from reviewer 3. Figure 7 is a little under annotated. Its not clear what the interval is between points. The interval shown into the panels to too large to allow tracking. I think its needs to be at least every 4 seconds. They should provide the interval and add time in seconds to the plots. I think the authors can improve the presentation to better show off all the work they have done. I suspect that revision is a little rushed and just needs polish, which they can provide.

Reviewers' comments:

Reviewer #1 (Remarks to the Author):

None

Reviewer #2 (Remarks to the Author):

The authors have addressed my concerns and those of other reviewers. The authors have made significant refinements and the general appearance that vesicles appearance corresponded to signal termination in micro clusters are less striking after the revisions and the role of Calcium will be pursued in future studies. I think this is reasonable given the amount of data in the current manuscript. They have also attempted to extend the LLS studies using additional data they had in hand, which improve the depth of analysis. They have also undertaken a significant manual segmentation of EM data to better demarcate different types of membrane structures. The authors also performed a more detailed analysis of the LAT flashes. I accept the authors arguments that the issues are still contentious and that the analysis here will further clarify the trafficking of lat in time frames of second and minutes following triggering of TCR signaling.

We thank the Reviewer for supporting our publication.

Reviewer #3 (Remarks to the Author):

General comment:

One of the main comment from the three reviewers was that it would be important to analyze the extent of T cell activation that occurs during the first 30-60 s of activation prior to relocalization of the pool of vesicular LAT at the IS in order to relate that to the 2 phases of T cell activation the authors are eluding to in their title.

Although the authors have made new experiments to answer this point, the results of the experiments shown in the paper do not allow to draw the conclusions made by the authors.

We have addressed this Reviewer's comments to the best of our ability below.

Specific comments:

Indeed, the microscopy analysis of the recruitment of signaling molecules presented in Figure 7, only show one cell for each signaling molecule studied. Several images should be analyzed and quantified rigorously to draw any conclusion. It is particularly important since it seems that at t_0 Grb2 and SLP76 are already associated with Lat, which raises the following question "is t_0 really a t_0 ?" and thus the all problem of the kinetic.

In all our figures, we present t_0 as the earliest observable time-point at the start of imaging at the microscope. From the footprint at t_0 of the cells shown in the revised version of Figure 7A-C, these cells are very early in the activation process (see Barr et al., 2016 JCS 129 Supp Fig 1 for classification of spreading T cells). In these panels of Figure 7, we have quantified multiple

microclusters at different times in activation. At early time points, line scans through microclusters show absence of VAMP7 signal above background, and coincident peak fluorescence intensities of LAT and the associated effector protein, thus confirming colocalization of microcluster components prior to vesicle recruitment. At later time-points line scans through microclusters show coincident peak fluorescence intensities of VAMP7, LAT and the associated effector protein, demonstrating the continued colocalization of microcluster components upon vesicle recruitment.

In response to the Reviewers comment that Grb2 is already associated with LAT, we focused on microclusters that formed during imaging to get a true t_0 for an individual microcluster. We evaluated the recruitment of LAT and Grb2 to the forming microcluster. We observed that LAT and Grb2 signals increased coincidentally as the microcluster formed, indicating that both LAT and Grb2 are recruited simultaneously to the microcluster. We show this in two examples presented in Figure 7D and E.

It is also very striking that VAMP7 in blue in the figure is seen in the early time point (48 sec), which somehow contradicts what the authors wrote about the kinetic of recruitment of the VAMP7-positive vesicular pool of Lat.

In Figure 2 we show that there is a range of times for recruitment of VAMP7 vesicles to the immune synapse (26sec to 129 sec). This range is probably due to the stochastic nature of MTOC location relative to the stimulatory surface in a particular cell as it drops onto the stimulatory coverslip. Therefore the recruitment of VAMP7 vesicles at 48sec is consistent with what has been described in an earlier figure in the paper.

Finally the figure as it is, is difficult to interpret since time scale is not clearly indicated in 7D, E and F (Does t_{14} correspond to 14sec? What is the scale in total fluorescence intensity representation in right panels?).

We appreciate the reviewer's comment and have now annotated Figure 7D and E with time corresponding to seconds, with t_0 being the initiation of imaging. We have also indicated that the time montage shown in the figure corresponds to 3sec/frame. These times are also indicated in the graph to the right. In the figure legend we have indicated that RFI corresponds to relative fluorescence intensity normalized from 0 to 1 for the time montage shown on the left.

For the WB analysis of the signaling in activated Jurkat T cells, the authors should also performed a quantification of several experiments to present a statistical analysis of the kinetic of activation. It is particularly important since activation pattern and kinetic seems variable in the 2 experiments presented in the figure: compare DMSO conditions in S1C and S1D for phospho-SLP76 and phospho-PLC γ 1. It is also striking that little phosphorylation of LAT is observed before 30 seconds in figure S1C (quantification shows a 3.3 increase at 10sec and 2.8 at 30 seconds for phospho-Lat but this seems overestimated when compared to blot below

showing 2.9 and 2.4 increase for phospho-SLP76 at the same time) and D (No increase of phospho-LAT at 10 and 15 seconds, 4.5 fold at 2 min). This is in contradiction with the fact that the plasma membrane pool of LAT would mostly contribute to activation. The peak at 2 minutes may indeed correspond to the recruitment of the vesicular pool suggesting that signaling requires recruitment of vesicular pool.

That LAT can be phosphorylated within seconds of TCR engagement is well established. A publication from our lab has previously shown that though LAT phosphorylation peaks at 2 minutes, it can be detected as early as 10 sec (Houtman et al., JI 2005). In our hands, we have detected LAT and SLP-76 phosphorylation by western blotting as quickly as 5 seconds after T cell activation (Figure 1A Response to Reviewers). So we do not agree with this reviewer's comment that the kinetics of phospho-LAT activation reflects a requirement for vesicular pool recruitment at later times.

However, in response to the Reviewer's comments about quantification, we quantified 3 independent experiments in which we compared DMSO treated control with BAPTA/nocodazole treated cells. The results of the quantification are shown in Figure 1B and C Response to Reviewers. Though we clearly do see the trends described in the imaging data presented in Figure 1 (BAPTA results in increased signaling at later time-points and nocodazole results in decreased signaling at later times), the variation between experiments resulted in large error bars. However, we do not think the biochemistry is required to support the imaging data presented in Figure 1, which stands on its own to show that microclusters at early times are unaffected by either BAPTA or nocodazole treatment. So we have now removed the representative set of blots that were originally included in Fig. S1.

Note that the experiments in Figure 1 were done to address certain predictions of models proposed by Soares et al. 2013. Our data indicated that these models are incomplete and we moved to the latest microscopy techniques to demonstrate that early signaling precedes vesicular recruitment. We also note that the data in Figures 2-4 and 7 unequivocally make the case for LAT activation prior to vesicle recruitment.

Concerning the results obtained with BAPTA and nocodazole they are difficult to interpret:
- For Western blots, because of a lack of quantification of several experiments and of the variability of the 2 experiments presented. No effect of nocodazole early on because no early activation of phospho-Lat and phospho-PLC γ in DMSO but clear effect on phospho-SLP76 as early as 10 sec in the one experiment presented. It also seems that there is less phospho-LAT at 2 minutes in figure 1E. Thus, the conclusion of the authors that nocodazole does not affect early signaling in T cells is in contradiction with the results they show.

Please see comments above.

- For microscopy images, quantification of images only for phospho-Lat and not for phospho-Tyr (which would give a more global idea of signaling in T cells). Moreover, it seems that the distribution of dots in 1B and perhaps 1C is not normal (bimodal). Is the statistical analysis used

appropriate for such a distribution? Although authors show phospho-Lat and total Lat in 1D they do not quantify total LAT, which would also be important to document.

In response to the Reviewers' comment, we have now included quantification of pY clusters and total LAT (LAT-ruby) clusters in Figure S1B and D. Both sets of results are consistent with the phospho-LAT quantification presented in Figure 1.

For 1B and C we used the standard unpaired t test using Graphpad prism. As part of the t test analysis, Prism uses an F test to compare the variance of the two groups. For datasets shown in Figures 1B and C, the P value for variance was not significant, thus confirming that it is valid to use the unpaired t-test.

- The interpretation of the results on BAPTA by the authors is not stated clearly enough. The results indeed clearly rule out the fact that exocytosis of the vesicles containing Lat at the immune synapse is required for phosphorylation and recruitment of LAT at 5 minutes. Thus the docking model of LAT vesicles just below the plasma membrane without fusion is the correct model.

[Redacted]

We are reluctant to overinterpret the data using this pharmacological agent and we believe that further addressing the docking model is outside the scope of this paper. However, we have now referred to the docking model in our Discussion section (pages 23 and 24).

Figure 1 Response to Reviewers

Figure 1 Response to Reviewers: Biochemistry of T cell activation. A. Jurkat cells were stimulated with CD3 in solution and lysed at the indicated time-points. Cellular proteins were separated by PAGE and phosphorylation of specific sites on LAT and SLP-76 determined by immunoblotting. B and C. Jurkat cells were treated with DMSO or nocodazole, stimulated with CD3 in solution and lysed at the indicated time-points. Cellular proteins were separated by PAGE and phosphorylation of specific sites on LAT, SLP-76, PLC γ 1 was determined by immunoblotting. Graphs represent quantitation of the ratio of pLAT, pSLP-76 and pPLC γ 1 to total actin at the indicated times. Data are means \pm SEM of three independent experiments.